# T cell microvilli constitute immunological synaptosomes that carry messages to antigen-presenting cells

Hye-Ran Kim[1,2], YeVin Mun[1,2], Kyung-Sik Lee[1,2], Yoo-Jin Park[1,2], Jeong-Su Park[1,2], Jin-Hwa Park[1,2], Bu-Nam Jeon[1,2], Chang-Hyun Kim[1,2], Youngsoo Jun[1], Young-Min Hyun [3], Minsoo Kim[4], Sang-Myeong Lee[5], Chul-Seung Park[1], Sin-Hyeog Im [6,7] & Chang-Duk Jun[1,2]

Microvilli on T cells have been proposed to survey surfaces of antigen-presenting cells (APC) or facilitate adhesion under flow; however, whether they serve essential functions during T cell activation remains unclear. Here we show that antigen-specific T cells deposit membrane particles derived from microvilli onto the surface of cognate antigen-bearing APCs. Microvilli carry T cell receptors (TCR) at all stages of T cell activation and are released as large TCR-enriched, T cell microvilli particles (TMP) in a process of trogocytosis. These microvilli exclusively contain protein arrestin-domain-containing protein 1, which is directly involved in membrane budding and, in combination with vacuolar protein-sorting-associated protein 4, transforms large TMPs into smaller, exosome-sized TMPs. Notably, TMPs from CD4$^+$ T cells are enriched with LFA-2/CD2 and various cytokines involved in activating dendritic cells. Collectively, these results demonstrate that T cell microvilli constitute "immunological synaptosomes" that carry T cell messages to APCs.

[1] School of Life Sciences, Gwangju Institute of Science and Technology (GIST), Gwangju 61005, Korea. [2] Immune Synapse and Cell Therapy Research Center, Gwangju Institute of Science and Technology (GIST), Gwangju 61005, Korea. [3] Department of Anatomy, Yonsei University College of Medicine, Seoul 03722, Korea. [4] Department of Microbiology and Immunology, David H. Smith Center for Vaccine Biology and Immunology, University of Rochester, Rochester, New York 14642, USA. [5] Division of Biotechnology, College of Environmental and Bioresource Sciences, Chonbuk National University, Iksan 54596, Korea. [6] Academy of Immunology and Microbiology (AIM), Institute for Basic Science (IBS), Pohang University of Science and Technology, Pohang 37673, Korea. [7] Division of Integrative Biosciences and Biotechnology (IBB), Pohang University of Science and Technology, Pohang 37673, Korea. Correspondence and requests for materials should be addressed to C.-D.J. (email: cdjun@gist.ac.kr)

An extensive body of evidence indicates that surface proteins are commonly transferred between immune cells in vitro and in vivo, and it is clear that this phenomenon is widespread. With characteristics distinct from enzymatic cleavage or exosome-mediated transfer, such cell-surface protein transfer has been referred to by different investigators as "absorption"[1], "internalization"[2], or "trogocytosis"[3,4] (from the Greek "trogo", meaning to gnaw or nibble). In general, the process of trogocytosis is antigen-dependent[5], with one of the best-known examples being the transfer of peptide-major histocompatibility complexes (MHCs) from antigen-presenting cells (APCs) to T cells[2,6]. Bidirectional trogocytosis has also been described, in which mouse APCs acquire the T cell receptor (TCR), and the counterpart T cells acquire peptide-bound MHCs[7]; however, the molecular mechanism underlying the process of trogocytosis remains unclear. Proposed mechanisms include endocytosis of the two synaptic membranes or an enclosed membrane being "torn off" as cells part.

As a separate field of study from trogocytosis, "direct vesicle budding" from the plasma membrane has been demonstrated, with the resulting vesicles termed microparticles, shedding vesicles, ectosomes, or arrestin-domain-containing protein-1 (Arrdc1)-mediated microvesicles[4]. Unlike trogocytosis, direct vesicle budding requires enzymes, including the ATPase vacuolar protein-sorting-associated protein 4 (Vps4) and the endosomal-sorting complex required for transport-1 component tumor-susceptibility gene 101 (TSG101)[8,9]. Arrdc1 is exclusively localized to the plasma membrane and mediates microvesicle budding directly from the plasma membrane by co-opting Vps4 and TSG101[8,9]. Consistent with this finding, a recent report demonstrated that TSG101 sorts TCRs into the immunological synapse (IS) center, and thereafter Vps4 mediates membrane budding of the TCR-enriched microvesicles[10]. However, the report was unclear regarding the exact location in the plasma membrane from which the microvesicles budded[10].

Lymphocytes contain abundant flexible projections termed microvilli, the inside of which contain many parallel bundles of actin filaments that extend the cell membrane in the form of a finger. Previous reports demonstrated microvilli involvement during the initial rolling phase of extravasation via segregation of two lymphocyte receptors, L-selectin, and α4β7 integrin on the tip of the microvilli, which is critical for lymphocyte attachment and rolling under physiologic flow[11]. Since then, the biological role of microvilli has remained surprisingly unrecognized. However, a recent super-resolution microscopy approach demonstrated that TCRs are highly clustered on the tips of microvilli, immediately highlighting these surface projections as effective sensors for antigenic moieties on APCs or target cells[12]. Shortly thereafter, the mechanism of how microvilli on T cells search opposing cells and surfaces before and during antigen recognition was described[13]. However, in addition to their key role in antigen sensing on APCs, whether they serve essential functions during T cell activation remains largely unclear.

In this study, we observe that single T cell contacts with APCs occur through microvillar extensions, which appear to serve as locations for sequestration of immunologically important molecules, including TCR complexes, costimulatory and adhesion molecules, and various cytokines. We find that microvilli are separated from the T cell body by the combined action of two independent mechanisms (trogocytosis and membrane budding) and are deposited at the surface of cognate APCs, thereby potentially acting as an effective means of delivering T cell messages to cognate APCs. Consistent with this potential role, these T cell microvilli-derived particles (TMP) are independently capable of activating cognate dendritic cells (DC). Therefore, our findings suggest that T cell microvilli might serve as "immunological synaptosomes" with TMPs as a class of membrane vesicles serving as conveyors of T cell messages or traits to cognate APCs.

## Results

**T cells generate microvilli particles upon TCR stimulation.** As shown in Fig. 1a (panel i), T cells contain abundant finger-like microvilli on their surface, although their roles and destinies are unclear. At the very early stage of IS formation, microvilli are polarized toward the surface of antigen-bearing APCs (Fig. 1a, panel ii; ~5 min), presumably to scan and sense the antigens on APCs[12,13,15]. Similarly, when naive T cells are placed on anti-CD3 antibody-coated coverglass, these microvilli extend to the antibody-coated surface (Fig. 1b, early). However, the microvilli mostly disappear as the IS matures (Fig. 1b, panel ii; ~30 min; mature), during which large-scale actin rearrangement occurs in the distal supramolecular activation cluster (dSMAC) of the IS[16]. This phenomenon might have impeded recognition of the importance of microvilli in the activation process of T cells. We noticed that some microvilli remain on the outermost edge of mature T cells and are more evident at the late stage (Fig. 1b, mature and late). Strikingly, many rod-shaped microvilli were separated from the T cell body and spread on the antibody-coated surface during the terminal stage (Fig. 1b, terminal). Microvilli separation is not an artefact, as this phenomenon also occurs on planar lipid bilayers presenting peptide-MHC and intercellular adhesion molecule-1 (ICAM-1) (Fig. 1c). Transmission electron microscopy (TEM) showed that separated microvilli were irregular in size, with some comparable to that of mitochondria (Fig. 1d). We named these particles TMPs (Fig. 1d).

Because EM analysis is limited in its ability to represent microvilli dynamics during IS formation and dissociation, we developed a fluorescence-based method to trace microvilli movement under live conditions. To that end, we selected tetraspanin protein 25 (TSPAN25)[17] and a single transmembrane protein V-set and transmembrane-domain-containing 5 (Vstm5)[18]. Both proteins were previously identified as localizing at membrane-protrusive regions, especially in microvilli or filopodia[17,18]. We observed that TSPAN25 was highly expressed in T cells, and that green fluorescent protein (GFP)-fused TSPAN25 (TS25G) overlapped with cortical F-actin in T cells (Supplementary Figs. 1a and b). However, TSPAN25 overexpression increased the number and length of microvilli in 293 T cells (Supplementary Fig. 1b), as well as the thickness of microvilli in Jurkat T cells (Supplementary Fig. 1c). By contrast, although Vstm5 was expressed at relatively low levels in T cells, GFP-fused Vstm5 (V5G) minimally affected microvilli length as compared with TSPAN25 (Supplementary Fig. 1b and c). Additionally, both wild type and V5G were recovered in the membrane fraction and not cleaved by Endo H (Supplementary Fig. 1d), suggesting that Vstm5 is a membrane protein. Nevertheless, in contrast to molecules that were only segregated at the tip of microvilli, such as TCRβ and ζ-chain in this study, Vstm5 displayed no preferential clustering at the tip of microvilli, but rather was distributed throughout the entire stalk region of microvilli, expressed at low levels in the inner area of cells (Fig. 2a), and significantly co-localized with cortical F-actin (Fig. 2b). Vstm5 contained a potential palmitoylation motif (CXXC) at the end of its transmembrane sequence[18], which was confirmed by [3]H-palmitate labeling (Supplementary Fig. 1e). Accordingly, deletion of the CXXC motif (ΔN4) reduced microvilli localization in both 293 T and Jurkat T cells (Supplementary Fig. 1f), suggesting a critical role of palmitoylation for Vstm5 localization at microvilli.

Vstm5 had minimal effects on T cell activation (Supplementary Figs. 2a and b), presumably because proteins with low expression normally would not be expected to significantly alter cellular

functions. We also generated a CRISPR/Cas9-based *Vstm5*-knockout mouse (*Vstm5*$^{-/-}$; Supplementary Figs. 2c and d) and found that *Vstm5*$^{-/-}$ T cells presented normal microvilli (Supplementary Fig. 2e), whereas the cells showed slightly reduced interleukin (IL)-2 production, but only when T cells were incubated with staphylococcal enterotoxin B (SEB)-loaded CD19$^+$ B cells (Supplementary Fig. 2f). Taken together, these results demonstrated that Vstm5 is a useful probe for microvilli study. Similar to the EM study, V5G signals reflected the presence of microvilli during all stages of T cell activation (Fig. 2c). At the bottom of late-stage T cells, V5G was significantly co-localized with F-actin (Fig. 2d), indicating that V5G$^+$ microvilli consisted

of actin bundles. In the terminal stage, we observed a number of rod- or round-shaped microvilli particles that were scattered around the cells (Fig. 2e, white box), and real-time separation of microvilli from the T cell body was clearly visible under total internal reflection fluorescence microscopy (TIRFM) (Fig. 2f and Supplementary Movie 1). Using V5G and, in some cases, TSPAN25, we determined how microvilli move during IS formation/deformation and release from the T cell body.

**Microvilli bring TCR clusters to the central (c)SMAC.** We initially determined the 3-dimensional dynamic movements of

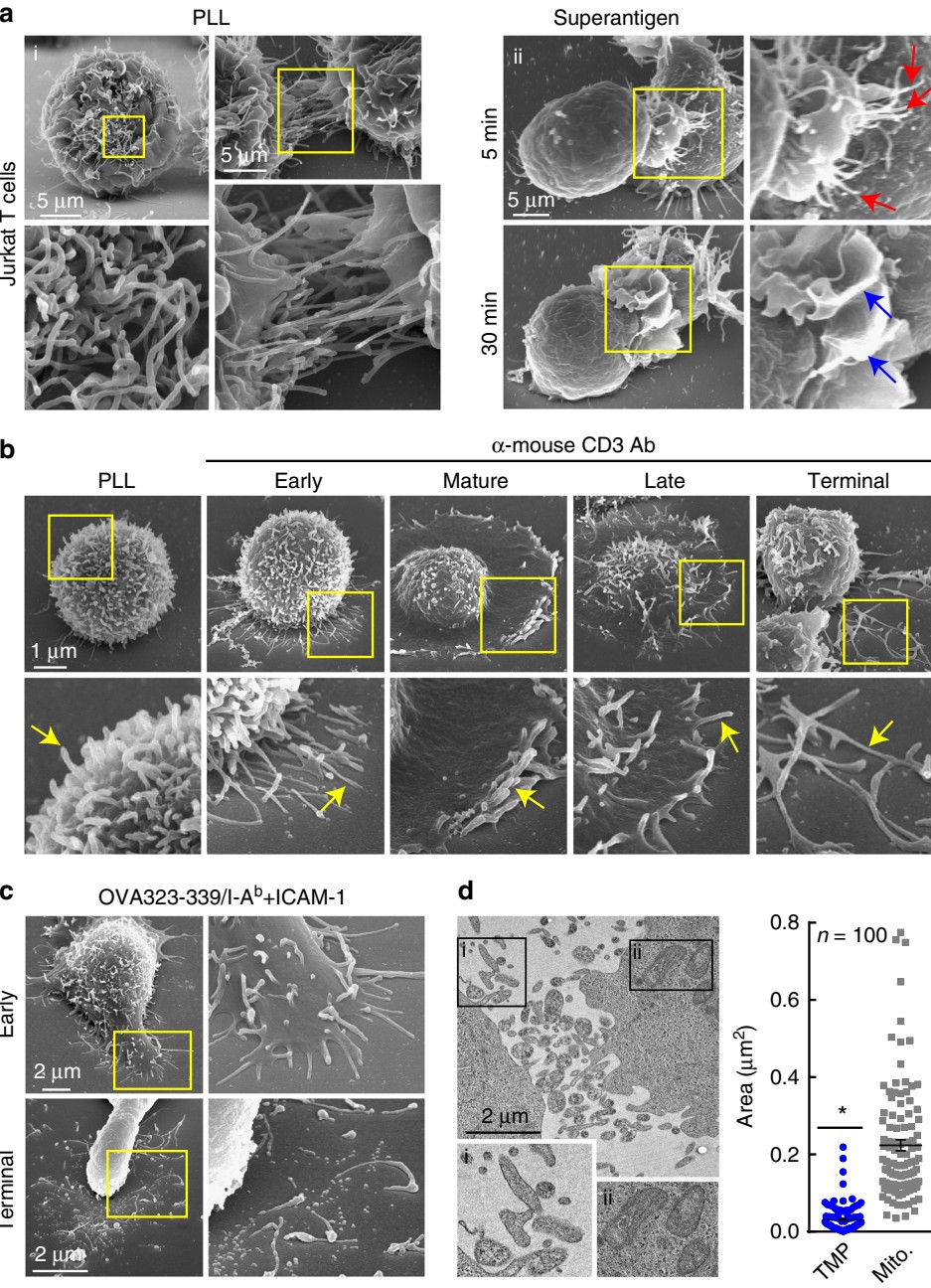

**Fig. 1** T cells generate microvillus-originated particles upon TCR stimulation. **a** T cell microvillus polarization toward antigen-bearing B cells at the early stage of IS. SEM of resting single (left) or two adjacent (right) Jurkat T cells on PLL (i) and co-incubated with SEE-loaded Raji B cells (5, 30 min) (ii). Arrows indicate microvilli (red) and membrane ruffles (blue). **b**, **c** SEM of OTII naive CD4$^+$ T cells on PLL with the anti-CD3 antibody (**b**) or OVA peptide/I-A$^b$ and ICAM-1 (**c**) at various time points. Early (early stage, 1 min), mature (1–5 min), late (10–20 min), and terminal stages (> 20 min). **d** Jurkat T cells were placed on anti-CD3-coated coverglass for 30 min and observed by TEM. The areas of TMPs and mitochondria were quantitated using ImageJ software. \*$P < 0.01$ vs. mitochondria

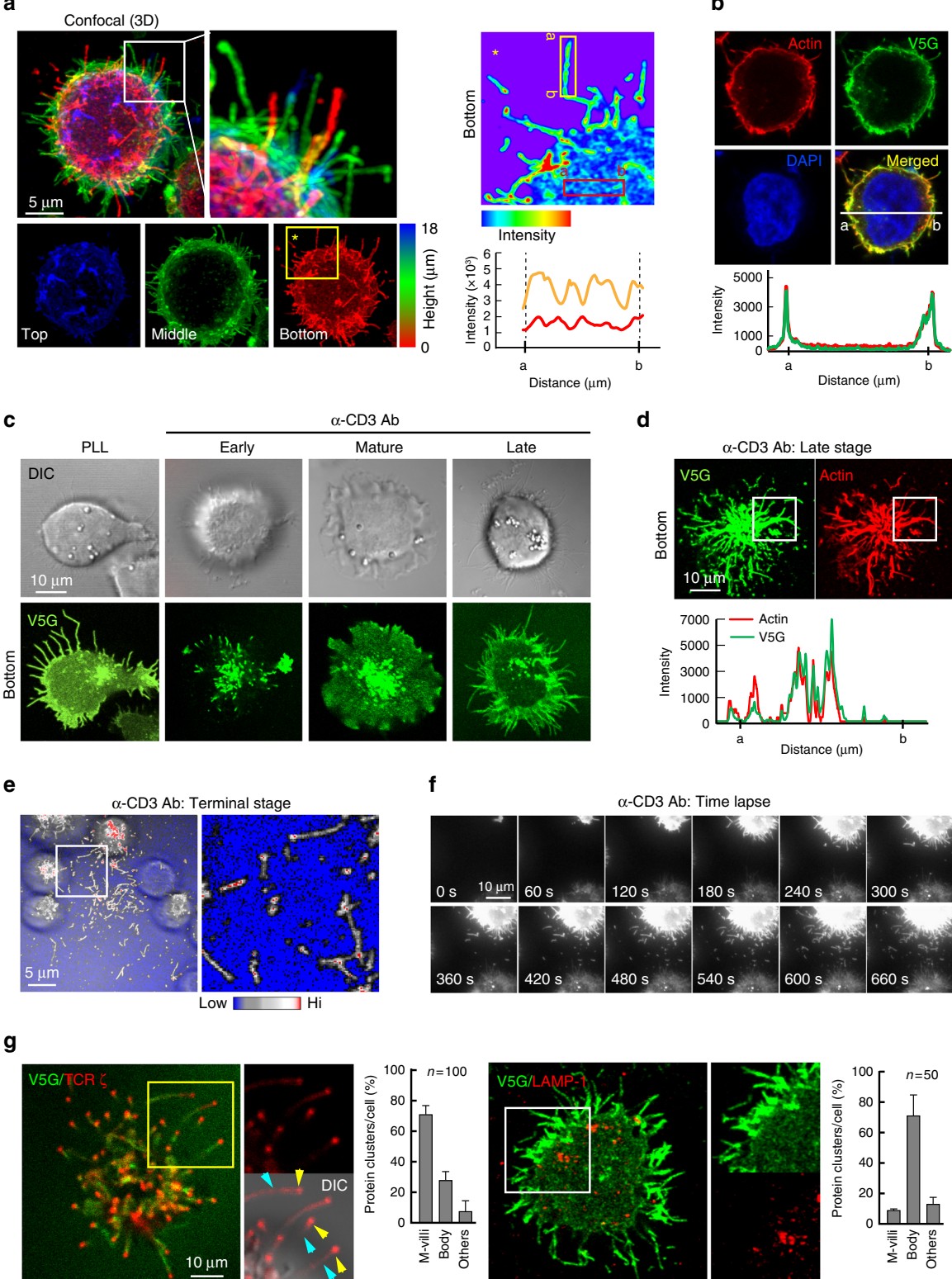

**Fig. 2** T cells generate V5G+ microvilli particles on anti-CD3-coated surfaces. **a** V5G+ Jurkat T cells were imaged by confocal microscopy and color-coded by z-position (left). Comparison of V5G fluorescence intensity between microvillus shaft and membrane surface (right). **b** Co-localization of V5G with F-actin. V5G+ Jurkat T cells on PLL were stained with TRITC-phalloidin. **c** Confocal images of V5G+ Jurkat T cells on anti-CD3-coated coverglass. Cells were monitored for 60 min, and representative images of each stage are shown. **d** Co-localization of V5G+ microvilli with F-actin. V5G+ Jurkat T cells on anti-CD3-coated coverglass and incubated for 20 min to 30 min were stained with TRITC-phalloidin. The fluorescence intensity profiles from **a** and **b** were analysed with ZEN v2.1 (for **a**, **b**, and **d**; Carl Zeiss). **e** Separated V5G+ microvilli particles are colored at the terminal stage. **f** Real-time V5G+ microvilli particles generation from activated T cells. V5G+ Jurkat T cells on anti-CD3-coated coverglass were observed by TIRFM (see also Supplementary Movie 1). **g** Jurkat T cells expressing V5G and TCRζ_ptdTomato were placed on anti-CD3-coated coverglass for 30 min. Protein localization at the terminal stage of T cell activation was determined. For LAMP-1 imaging, primary and secondary antibodies were used

microvilli during T cell interactions with antigen-loaded APCs or an anti-CD3-coated surface. Similar to the scanning EM (SEM) study, V5G[+] microvilli polarized toward the contact site of a T cell and a B cell at the early stage of IS and moved into the cSMAC at the mature stage (Supplementary Fig. 3a), Interestingly, at the mature stage, V5G signals were seen as puncta and highly co-localized with F-actin spots at the cSMAC (Supplementary Fig. 3b and Supplementary Movie 2), with these spots possibly representing actin bundles that are the main components of microvilli (Supplementary Fig. 3b, white arrowheads). These results suggested that microvilli do not collapse during IS maturation[19,20], but instead are scattered around the cells (Figs.1b–d 2e, f; Supplementary Movie 1) or moved into the cSMAC (Supplementary Figs. 3a and b). A potential mechanism for how microvilli disappeared from the dSMAC, moved toward the cSMAC with TCR clusters during IS formation, and reappeared after synapse breakage is depicted in Supplementary Fig. 3c.

Because a recent report demonstrated that TCRs are clustered at microvilli tips[12], we determined whether microvilli contain TCRs. TCR ζ-chain signals were clearly observed at the tip of V5G[+] microvilli during the late stage of activated T cells (Fig. 2g). By contrast, lysosomal-associated membrane protein (LAMP)-1, a late endosomal marker, was almost excluded from microvilli (Fig. 2g). These results logically suggested that microvilli separated from the T cell body might contain a considerable number of TCR complexes and allied proteins.

To overcome weaknesses associated with antibody based work, we adopted a lipid-bilayer system presenting I-A[b] MHC class II and ICAM-1 and hypothesized that if centralization of the TCR clusters depends upon the movement of microvilli, both TCRs and V5G (or TS25G) should be observed together on the lipid bilayers. To this end, *OTII TCR* CD4[+] T cells were transfected with V5G (or TS25G), and their movements were observed in combination with TCRβ under TIRFM. Both proteins were observed as puncta and moved similarly toward the cSMAC (Fig. 3a, Supplementary Movie 3, and Supplementary Fig. 4a). These processes were completely dependent upon the intact actin cytoskeletons, as disruption of surface microvilli by actin modulators reduced cSMAC formation (Fig. 3b).

**Increased TCR[+] TMPs release during T cell kinapse.** T cells engage in two modes of interaction with antigen-presenting surfaces: stable synapses and motile kinapses[21]. Interestingly, a recent report demonstrated that all major T cell subsets spend more time in the kinapse mode, and that durable interactions for priming do not require stable synapses[22]. Therefore, we determined which mode of T cells produce more TMPs. In the presence of the OVA323-339 peptide, most OTII CD4[+] T cells formed strong synapses (Supplementary Fig. 4b); however, cells in the synaptic period released relatively few TCRβ[+] TMPs (Fig. 3c). By contrast, interestingly, T cells in kinapses produced additional–TCRβ[+] TMPs behind them (Fig. 3c, d, Supplementary Fig. 4b and c, and Supplementary Movie 4; arrowheads). The particles were irregular in their forms and larger than exosomes (L-TMPs; diameter: 20–500 nm); however, EM analysis showed that regular- and small-exosome-sized TMPs (S-TMPs; diameter: 20–40 nm) were also present in the inner area of the T cell under the transition period between synapse and kinapse, and that this area was surrounded by L-TMPs (Fig. 3e). The existence of S-TMPs suggests that previously reported TCR-enriched microvesicles[10] were budded from microvilli, which accumulated at the cSMAC (Fig. 3a). However, the fact that additional L-TMPs were released from T cells in kinapses strongly demonstrated that the cSMAC is not a secretory domain of microvesicle release. Instead,

the microvillus itself provided a secretory microdomain for TMP productions. Additionally, generation of TMPs required adhesion, as they were only significantly generated on the anti-CD3-coated surface or planar lipid bilayers presenting peptide-MHC and ICAM-1 (Fig. 3f). These results suggested that TMP generation required TCR signaling, as well as receptor-mediated T cell adhesion on the surface of antigen-bearing APCs.

We then confirmed the generation of TMPs during T cell interaction with APCs. Microvillar projection is not easily observed in vivo, as microvilli on the surface of T cells are too thin and short. However, if a T cell truly contacts APCs via microvilli, we expected that the T cell would form multiple bridges with APCs due to the numerous surface microvilli, which differs from membrane nanotubes, only a few of which are observed between connected cells[23]. Here, viral transduction of V5G enabled us to monitor microvilli extending from the T cell surface. As expected, although a single OTII CD4[+] T cell bound mainly to one B cell or one DC in the presence of antigen (OVA323-339), this cell also formed multiple bridges with other surrounding B cells or DCs (Fig. 4a).

To determine how multiple bridges are formed and TMPs are generated under live conditions, CD4[+] T blast cells expressing V5G were incubated with SEB-loaded DCs and monitored under a live-cell imaging microscope. Time-lapse imaging unambiguously revealed that T cells formed multiple bridges on the surface of DCs during T cell movement (Fig. 4b, Supplementary Movies 5 and 6). Additionally, L-TMPs were obviously generated from the elongated T cell microvillar tails (Fig. 4b, panel (i)), with some DC particles detaching and moving along with the T cell microvillar tail (Fig. 4b, panel (ii), yellow arrowheads; and Supplementary Movie 6), possibly suggesting that such particles were derived from DC microvilli, which are known to preferentially present MHC class II molecules[24,25]. The same results were also obtained when OTII CD4[+] T cells interacted with OVA-loaded DCs or B cells (Fig. 4c). To determine whether the separation of V5G[+]-microvilli is integrin-dependent, we used T cells from *lymphocyte function-associated antigen-1 (LFA-1)*-knockout mice[26]. Because *LFA-1[−/−]* T cells significantly lose their ability to bind to ICAM-1[+] DCs, we also investigated the ability of *LFA-1[+/−]* T cells (heterozygotes) to generate V5G[+] MPs, as these cells retain partial binding capacity toward ICAM-1[+] DCs (Fig. 4d). However, neither *LFA-1[−/−]* nor *LFA-1[+/−]* T cells left residual V5G[+] TMPs, suggesting that the high-affinity state of LFA-1 is a prerequisite for optimal microvilli disconnection.

**L-TMPs are further fragmented by the budding complex.** The release of TMPs from the T cell body suggests that these particles might comprise a vector specialized for intercellular communication. However, the remaining question was how large TMPs can be and remain capable of transfer to interacting cells without size reduction, as some rod-shaped particles are larger than a few micrometers. Surprisingly, we found that V5G[+] microvilli were further fragmented during and after disconnection from T cells in kinapses (Fig. 5a). Additionally, SEM analysis revealed that S-TMPs were generated from microvillar stalks and tips of moving or spread T cells (Fig. 5b, c, cyan blue arrows), suggesting that another mechanism in addition to adhesion-dependent trogocytosis is involved in converting L-TMPs to S-TMPs. The intermediate forms between microvilli and microparticles were easily observed by SEM analysis of OTII CD4[+] T cells on planar lipid bilayers presenting OVA peptide/I-A[b] and ICAM-1 (Fig. 5b). Therefore, we determined the size of TMPs after purification, as depicted in Supplementary Fig. 5a. Surprisingly, the diameters of TMPs following purification ranged from 20 nm to 100 nm, which is comparable to the size of exosomes[4] and was similar to

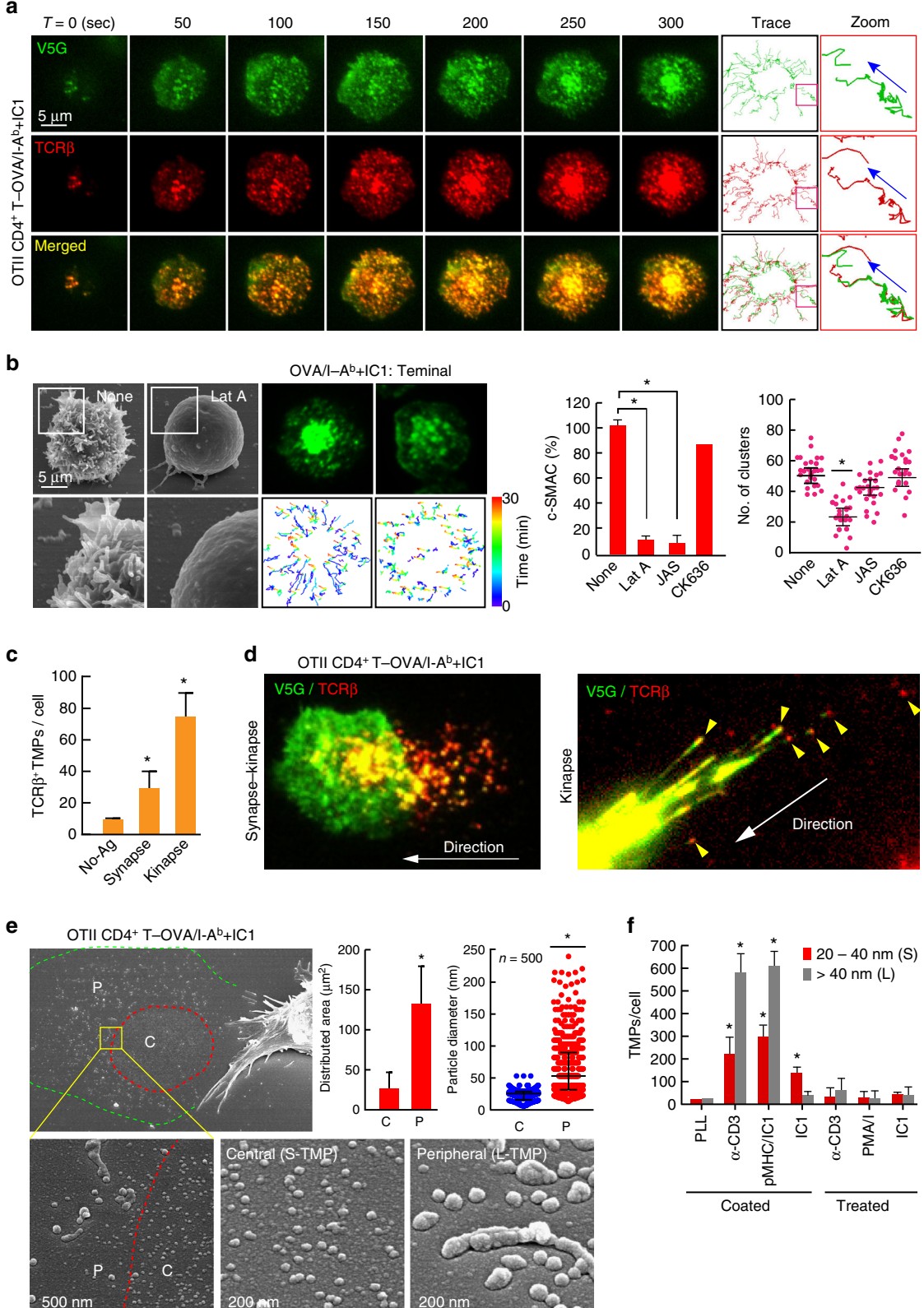

human and mouse T cells (Fig. 5d). Moreover, we observed that TMPs exhibited small and round-shaped morphologies (Fig. 5d). To identify proteins, purified TMPs were subjected to liquid chromatography tandem mass spectrometry (LC-MS/MS), with unsupervised hierarchical clustering analysis and principal component analysis (PCA) used to identify similarities and

differences in the whole proteome among total, vesicle, membrane, and TMP groups. The TMP and membrane groups clustered closely together but distal from the total and vesicle groups, indicating that the TMP group exhibited a proteomic pattern close to that of the membrane group (Fig. 6a). The proteins that were commonly detected in three independent experiments in the

**Fig. 3** TCR-enriched microvillus particles are released to a greater extent during T cell kinapse mode. **a** Co-localization of V5G with TCR microclusters. V5G[+] OTII CD4[+] T cells were stained with anti-TCRβ (H57Fab-Alexa594) and examined on a planar bilayer presenting OVA peptide/I-A[b] and ICAM-1 (see also Supplementary Movie 3). Relative time scales starting from the time of cell spreading are labeled on the images. Representative trajectory of individual TCR microclusters are shown in the right panel. **b** V5G[+] OTII CD4[+] T cells were pretreated for 30 min with (Lat A; 237 nM), JAS (100 nM), and CK636 (100 μM), followed by SEM (left) and TIRFM imaging to visualize V5G[+] microvillus movement on the lipid bilayer (middle). Boxed regions are shown enlarged at the bottom of the panels. Representative trajectories of individual V5G clusters (bottom). Statistical analyses of the frequency of cSMAC formation and the number of V5G clusters in OTII CD4[+] T cells (right). *P < 0.05 vs. none. Data represent the mean of three experiments ± SEM. **c, d** Release of TCR[+] TMPs during synapse and kinapse. V5G[+] OTII CD4[+] T cells were stained with anti-TCRβ (H57Fab-Alexa594) and observed on a lipid bilayer presenting OVA peptide/I-A[b] and ICAM-1 by TIRFM (see also Supplementary Movie 4). The numbers of TCRβ[+] TMPs were quantitated using ImageJ software (**c**). *P < 0.01 vs. no antigen (No-Ag). Data represent the mean of three experiments ± SEM. **e** L- and S-TMP release during synapse and kinapse. Cells from (**d**; synapse–kinapse) were subjected to SEM analysis. P, pSMAC; C, cSMAC. Particle diameters were analysed using 500 particles in the C and P of a single cell, and > 30 cells were examined. Data represent the mean of three experiments ± SEM. *P < 0.001 vs. C. **f** TMP generation from OTII CD4[+] T cells was examined in each condition. Cells were subjected to SEM, and the number of TMPs per cell was analysed using ImageJ software. Data represent the mean of three experiments ± SEM. *P < 0.001 vs. PLL

total, membrane, TMP, and vesicle groups were selected (Supplementary Data 1), and western blots were performed to identify the presence of T cell proteins identified by LC-MS/MS. Strikingly, TMPs exclusively contained a large amount of direct plasma-membrane budding (DPMB) complex components, such as Arrdc1, TSG101, and Vps4 a/b (Fig. 6b,c). Notably, Arrdc1 plays a pivotal role in DPMB, an intrinsic cellular process that underlies multivesicular-body independent vesicle formation at the cell surface[8]. Arrdc1 was highly concentrated at microvilli tips (Fig. 7a, arrowheads), and TSG101 was observed in the cytoplasm in multiple punctate foci, as is characteristic of endosome-associated localization[8]. However, TSG101 was redistributed to the microvilli when co-expressed with Arrdc1 (Fig. 7a). Interestingly, Arrdc1 was clearly co-localized with TCR clusters and moved toward the c-SMAC during IS maturation (Fig. 7b), followed by its release as a TMP upon T cell movement to form additional contacts on the lipid bilayer (Fig. 7c). Furthermore, the TCR ζ-chain was recovered with the Arrdc1-enriched fractions after ultracentrifugation of the sucrose gradient (Fig. 7d)[27]. Knockdown of either *Arrdc1* or *Vps4* showed little effect on the morphology of microvilli on the T cell surface (Fig. 7e), whereas many elongated microvilli tails or long rod-shaped microvilli particles were observed in T cells (Fig. 7f, g). Accordingly, TMP generation was significantly reduced (Fig. 7g), and TMPs were also significantly reduced in the absence of ICAM-1 on the lipid bilayer presenting OVA peptide/I-A[b] (Fig. 7h and Supplementary Fig. 6), suggesting that trogocytosis mediated by the LFA-1–ICAM-1 interaction, as well as membrane-budding reactions, were not independent processes, but cooperated to produce additional TMPs. Evaluation of the effects of various other antibodies, including anti-LFA1, anti-MHC, anti-CD43, and anti-CD44, revealed that none were involved in TMP generation (Fig. 7i), suggesting that TMP release was specifically associated with response to TCR activation.

**TMPs from CD4[+] T cells activate DCs independent of MHC.** Both proteomics and western blot analyses revealed that TMPs exclusively contained important T cell proteins, including the TCR complex (TCRα/β and CD3ε, γ, and ζ-chains) and costimulatory molecules, such as CD2 (LFA-2) and CD28 (Fig. 6b, c). These results implied that TMPs play a physiological role in cellular interactions. In agreement with this hypothesis, TMPs from OTII CD4[+] T cells (4-TMPs) deposited on the lipid bilayer strongly induced sustained calcium response in DCs (Fig. 8a and Supplementary Fig. 7a–c); however, we observed that calcium signaling was initiated, even when DCs were not loaded with OVA peptide, suggesting that DC activation by 4-TMPs is independent of cognate TCR–MHC signaling. To rule out the influence of OVA peptide remaining on the lipid bilayer, we

examined whether 4-TMPs purified from the anti-CD3-coated surface also evoked the same physiological DC responses as 4-TMPs on the lipid bilayer. To this end, 4-TMPs purified from the anti-CD3-coated surface (Supplementary Fig. 5a) were incubated with DCs in the presence or absence of SEB. The calcium response was observed regardless of SEB (Fig. 8b), indicating that the effect was independent of cognate TCR–MHC engagement. Moreover, DCs from Toll-like receptor (TLR)4-knockout ($TLR4^{-/-}$) mice did not respond to lipopolysaccharide (LPS) stimulation, but their calcium levels were significantly upregulated by 4-TMPs, demonstrating that the effect of 4-TMPs was independent of LPS signaling (Fig. 8b). Additionally, 4-TMPs induced the expression of costimulatory molecules, such as CD40, CD80, and CD86, in DCs (Fig. 8c, d). In general, 4-TMPs collected from T cells were sufficient to activate DCs at a ratio of 1:1 (T cell:DC) (Fig. 8c). Furthermore, blocking the Fc gamma receptor (FcγR) had little effect on DC activation, suggesting no influence by anti-CD3 antibodies used to purify 4-TMPs (Fig. 8d).

We then determined whether TMPs are important in the selective activation of physically interacting (or cognate) APCs or whether they are involved in the activation of bystander APCs. We employed a transwell chamber with a 0.4-μm pore size that allows TMPs but not T cells and DCs. OTII CD4[+] T cells and DCs were incubated in the absence (−) or presence (+) of OVA323-339 (Fig. 9a). Because noncovalent protein-protein interactions, such as antibody binding, can be dissociated by acid buffer treatment[28], we measured the surface transfer of T cell microvillar proteins, including TCR and V5G, on DCs after acid washing (Supplementary Figs. 8 and 9a). V5G[+] and TCR[+] signals were only observed in DCs directly incubated with OTII CD4[+] T cells in the presence of OVA peptide (Fig. 9a), suggesting that 4-TMPs released by peptide-specific CD4[+] T cells stably bind only to the surface of cognate antigen-bearing DCs. Consistently, the upregulation of costimulatory molecules was observed in V5G[+] or TCR[+] DCs (Fig. 9a), with similar results obtained using antibody purified 4-TMPs (Fig. 9b). These results strongly demonstrated that physical contact between TMPs and the DC surface was critical for TMP-mediated DC activation. To validate in vivo evidence of antigen-dependent TMP transfer (or TCRβ[+] particles), we injected either OVA257-264 (MHC class I) or OVA323-339 (MHC class II) peptides into *OTII TCR* transgenic mice and evaluated the expression of TCRβ[+] on the surface of DCs. Significant levels of TCRβ were only observed on DCs from OVA323-339- but not OVA257-264-injected mice, demonstrating that TMPs were transferred into the cognate DCs in vivo (Fig. 9c).

Along with the transfer of TCRs, we also determined the transfer of other membrane T cell proteins onto the surface of

DCs. We observed a significant increase in T cell proteins, such as CD4, CD28, CD2, and CD25, but not CD40L, on the surface of DCs (Fig. 10a). Because costimulatory proteins on the T cell surface induce signals that affect APC activation and/or maturation[29,30], we further determined whether some of the costimulatory proteins observed by LC-MS/MS analysis contributed to 4-TMP-mediated DC activation. To this end, we employed blocking antibodies that interfere with CD2/CD58, CD40L/CD40, and LFA-1/ICAM-1 interactions. Antibodies against CD2 and LFA-1, but not CD40L (present at low levels in TMPs), clearly reduced DC activation (Fig. 10b); however, the lack of complete inhibition of calcium signaling suggested that several factors in TMPs might be involved in the activation of

DCs. Because some cytokines, such as tumor necrosis factor (TNF)-α[31] and IL-4[32], also activate DCs, we determined the levels of cytokines or secretory proteins in 4-TMPs and compared these with levels observed in total cell lysates. Cytokine array revealed that TMPs were enriched with various T cell cytokines (Fig. 10c), among which we determined the levels of IL-33[33], TNF-α[31], IL-4[32], and IL-7[34], which regulate DC activation or maturation. These results indicated that several factors in TMPs regulate the activation of DCs.

## Discussion

Although many reports have documented that immune cells can extract surface molecules through the IS to which they are

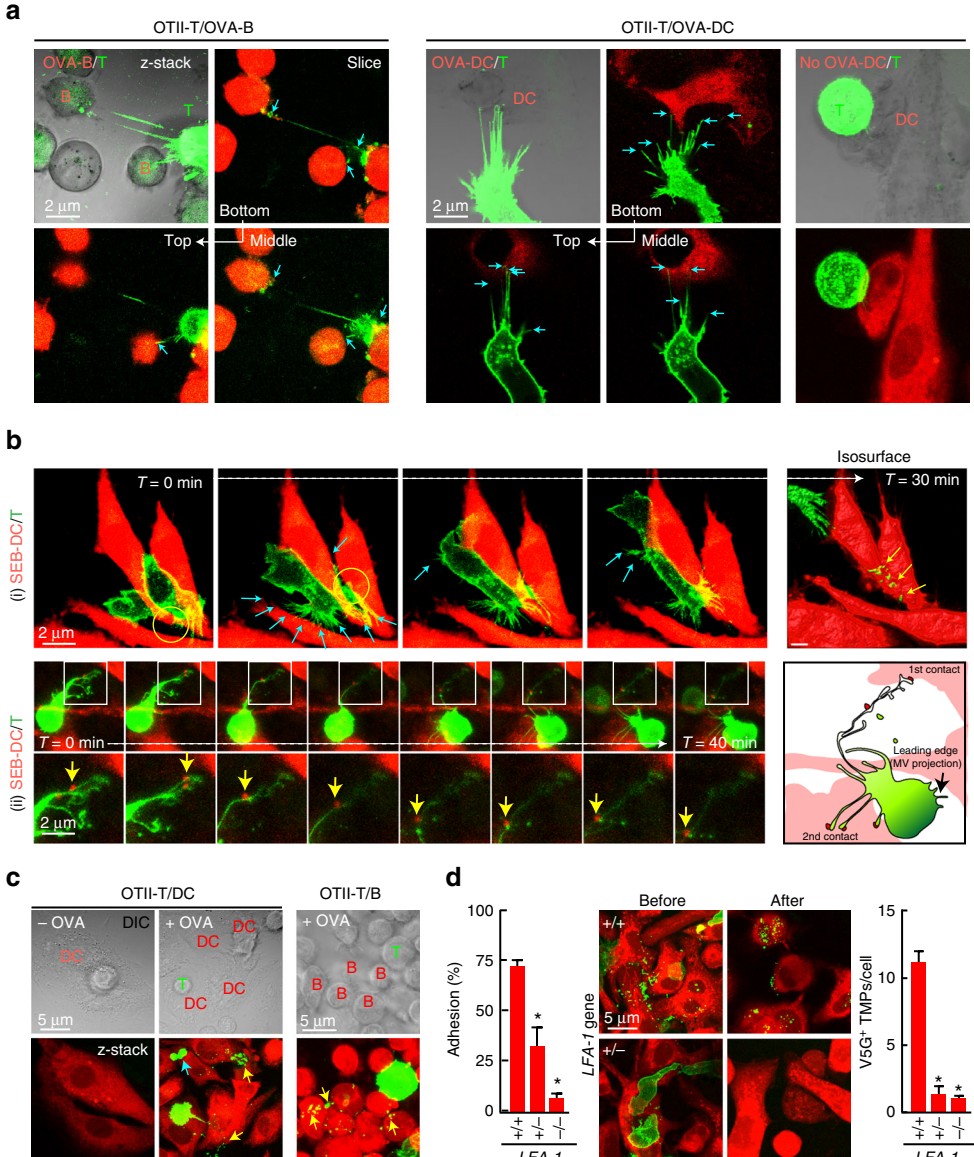

**Fig. 4** T cells leave TMPs on the surface of APCs in an integrin LFA-1-dependent manner. **a** Confocal microscopy of V5G⁺ OTII CD4⁺ T cells incubated with OVA323–339-loaded B cells or DCs for 30 min. Arrows indicate multiple contact sites. **b** Time-lapse imaging of V5G⁺-TMP generation from mouse T cells. CD4⁺ T cells were retrovirally transduced with V5G and incubated with SEB-loaded DCs for 30 min. Cyan arrows indicate the multiple contact sites. V5G⁺ TMPs (yellow arrows) on DCs in the last panel were visualized by isosurface rendering using Imaris software (see also Supplementary Movies 5 and 6). **c** V5G⁺ OTII CD4⁺ T cells from **a** were incubated for 1 h and observed by confocal microscopy. V5G⁺ TMPs are indicated by yellow arrows. **d** Adhesion of CD4⁺ T cells from wild type, LFA-1⁺/⁻, and LFA-1⁻/⁻ mice to mICAM-1-Fc-coated plates after anti-CD3/28 stimulation. Data represent the mean of three experiments ± SEM (left). Mouse CD4⁺ T cells from wild type, LFA-1⁺/⁻, and LFA-1⁻/⁻ mice were retrovirally transduced with V5G, incubated as described in **a**, and observed by confocal microscopy. The number of V5G⁺ TMPs per DC was quantified using Imaris software. Data represent the mean of three experiments ± SEM. *P < 0.01 vs. wild-type T cells

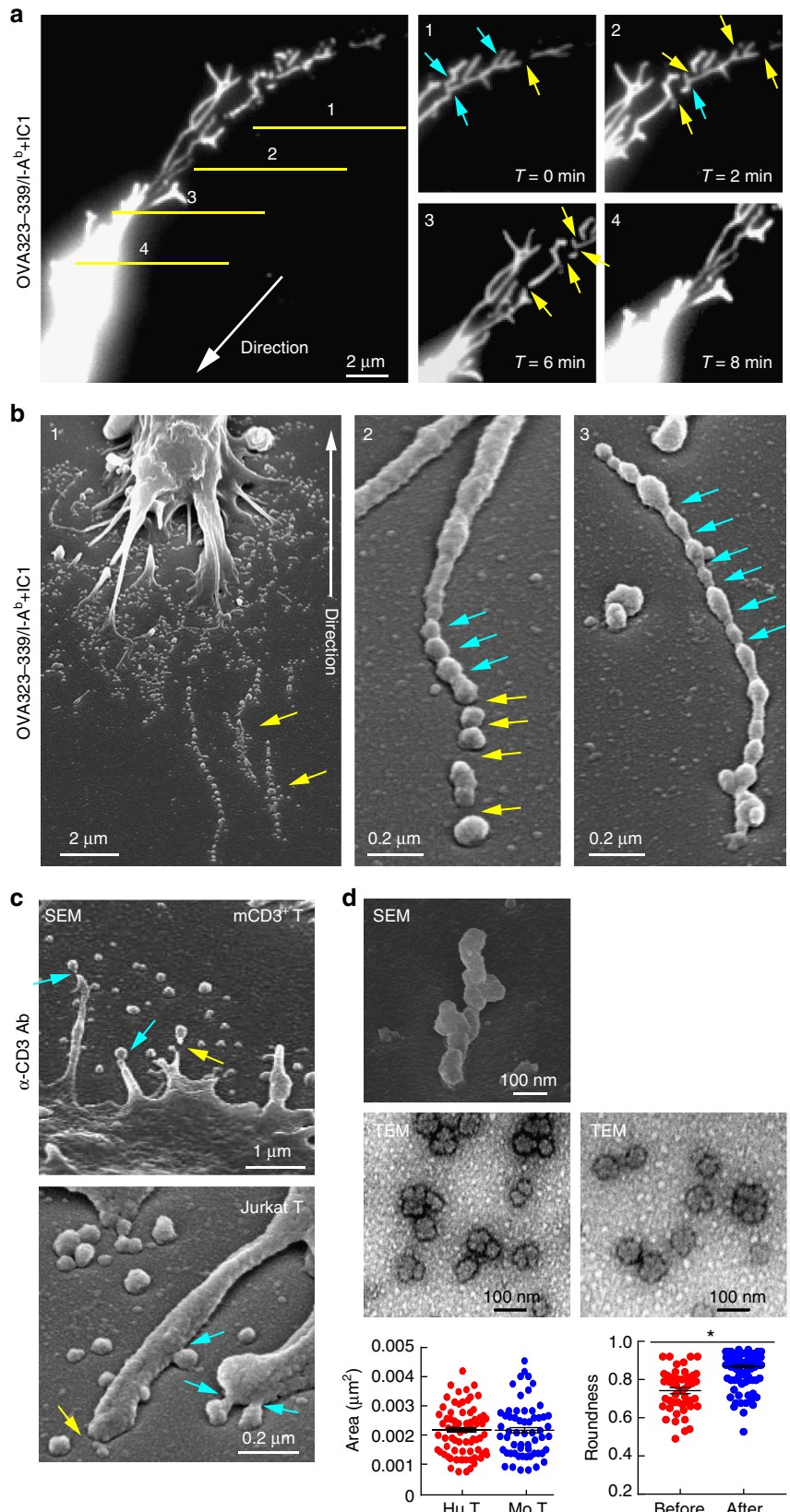

conjugated, the site of surface extraction, as well as the associated molecular mechanisms, remain unknown. In this study, we demonstrate that T cell microvilli are membrane organelles that are highly fragile to extraction when two cells separate. Surprisingly, membrane particles derived from T cell microvilli (i.e.,

TMPs) are enriched with many immunologically relevant proteins, including the TCR complex, and play a role as vectors to provide T cell messages or traits to physically interacting APCs.

Until now, the general importance of microvilli present on the surface of lymphocytes has been strongly underestimated, and

**Fig. 5** L-TMPs undergo morphological changes during and after disconnection from the cell body and are comparable in size to exosomes. **a** Fragmentation of V5G⁺ microvilli particles during and after disconnection from T cells in kinapse on a lipid bilayer (yellow arrows = fragmented sites; blue arrows = sites yet to be fragmented). **b** Microvillus-originated particles in the process of fragmentation. Cells from (**a**) were observed by SEM. Yellow arrows indicate the cleavage of L-TMPs into S-TMPs (1). Cyan blue arrows indicate the pre-cleavage sites in the microvilli (2) and L-TMPs (3). **c** SEM analysis of TMPs generated from CD3⁺ T cells on anti-CD3-coated coverglass. Yellow arrows indicate the cleaved TMPs from microvilli. Cyan blue arrows indicate TMPs budding from microvilli tips. **d** Mouse and human TMPs were harvested as described in Supplementary Fig. 5 and observed by SEM and TEM. The area and roundness of TMPs from human and mouse T cells were measured using ImageJ and Imaris software, respectively. Data represent the mean of three experiments ± SEM

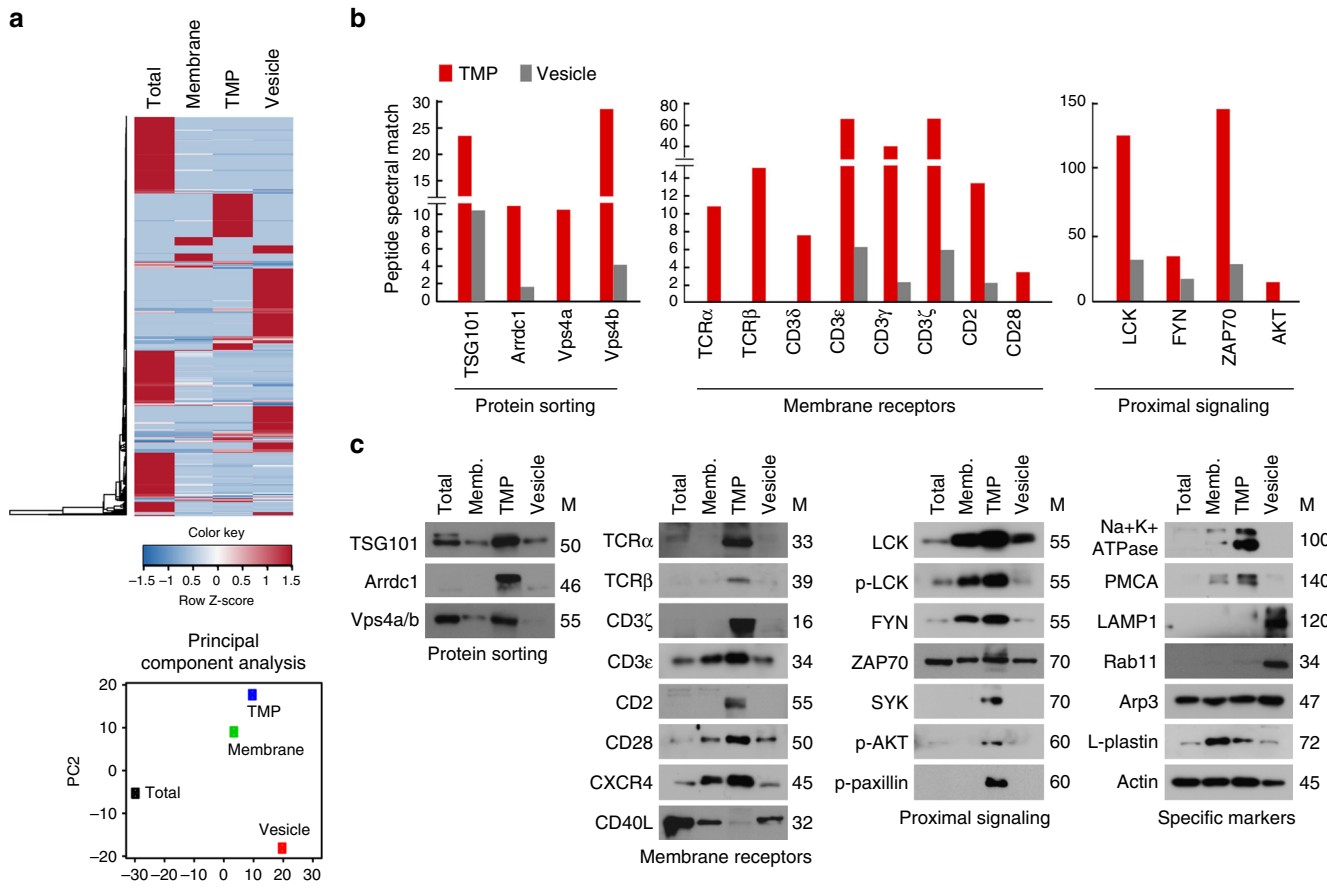

**Fig. 6** Identification of proteins in TMPs by LC-MS/MS. **a** Unsupervised clustering (upper) and PCA (bottom) of LC-MS/MS-based whole-proteome data from total, vesicle, membrane, and TMP groups. The heatmap represents relative expression patterns of proteins across the four groups, and hierarchical clustering of samples (column) and proteins (rows) was based on Pearson's correlation coefficient used to measure the distance. **b** Comparison of PSMs between TMPs and vesicles. **c** Western blot analyses of total lysates, membrane, TMPs, and vesicles from mouse CD3⁺ T cells

their essential functions remain unrecognized. We found that the early onset of T cell antigen recognition includes the extension of microvilli toward the cell that the T cell is recognizing. Using specific GFP-tagged probes, we monitored microvilli at all stages of T cell activation, findings that TCRs were segregated on microvilli tips, as recently reported[12], and also centralized to the cSMAC along with movement of the microvilli. We then found that large TCR-enriched TMPs (L-TMPs) were separated from the T cell body via trogocytosis and then scattered around the cSMAC or behind the moving T cells. Surprisingly, the particles were further fragmented by membrane-budding complexes, such as Arrdc1, TSG101, and vps4, which were also enriched at the microvilli tips, and formed typical exosome-sized S-TMPs. Immunologically important T cell molecules were selectively concentrated at the microvilli tips and in separated TMPs, which enabled TMP-mediated control of the activation state of

physically interacting DCs and suggested that TMPs constitute "immunological synaptosomes" that serve as a conveyors of T cell immunologic transmission.

A previous report demonstrated that extracellular membrane particles carrying the stem cell marker prominin-1 (CD133) are released from the microvilli of neural progenitors and epithelial cells[14,35]. Additionally, a recent report demonstrated that primary and motile cilia/flagella, which function as cellular antennae, also produce ectosomes[36], suggesting that membrane-protrusive regions, such as microvilli and cilia/flagella, could be the sources of membrane particles. However, generation of these membrane particles was not dependent upon cell–cell contact, but rather they were released through secretion via an unidentified mechanism. Therefore, the release of contact/adhesion-dependent TMPs is a unique and novel system and more highly related to trogocytosis than to vesicle secretion in activated T cells.

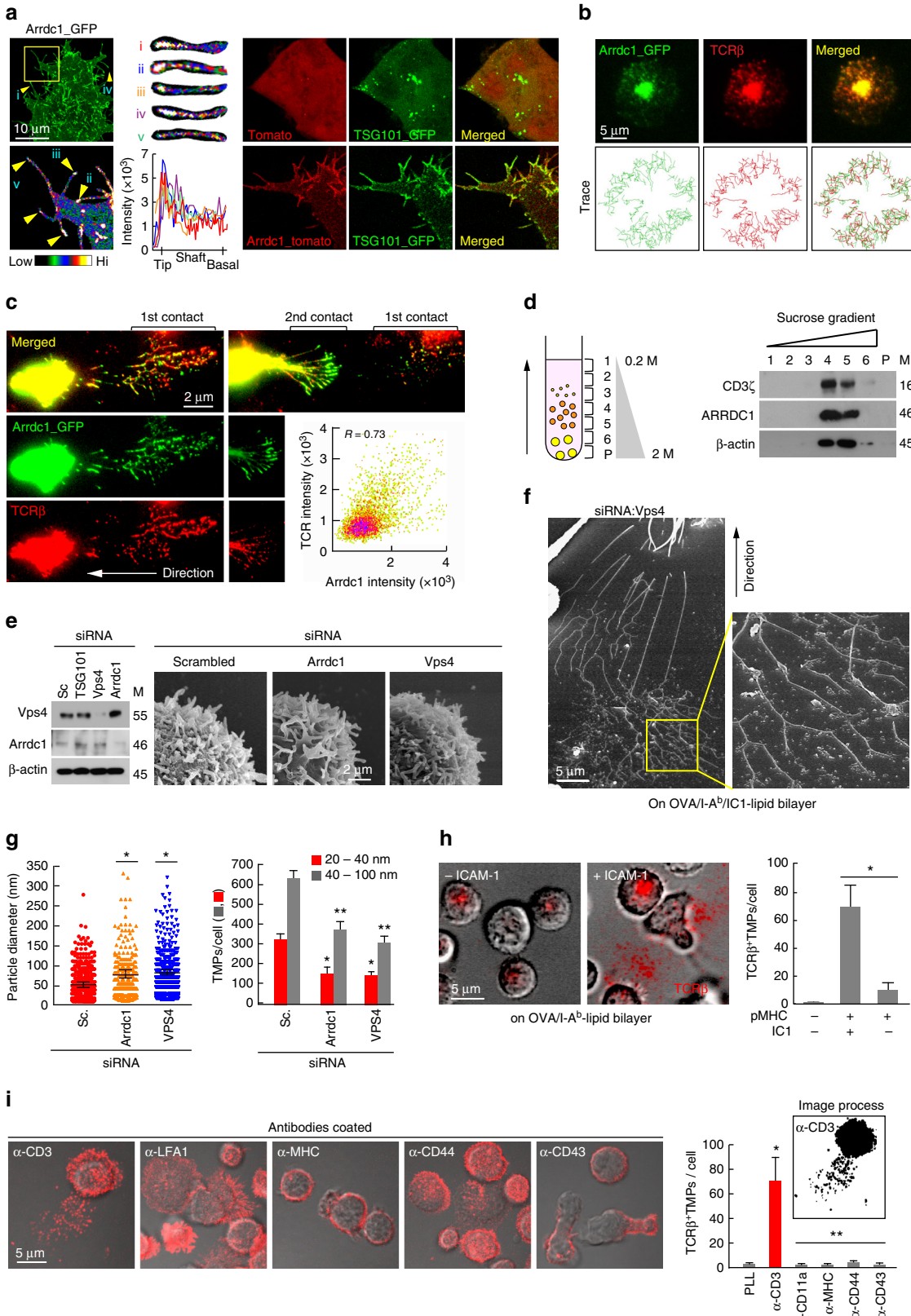

However, it is inappropriate to attribute this exclusively to trogocytosis, because the membrane-budding machinery is also involved in the cleavage of large microvillar fragments from S-TMPs. From this viewpoint, it would be interesting to compare the mechanisms associated with TMP generation with those of recently identified membrane vesicles, such as neutrophil trails[37] and migrasomes[38], as these two vesicles also require membrane contact with adhesion molecules or matrix in order to be generated[37,38]. Contact-dependent transfer of membrane particles has at least two merits. First, it can rapidly transfer the messages of donor cells to recipient cells. Second, this method can only transmit signals between physically interacting cells, thereby

**Fig. 7** L-TMPs are further fragmented by membrane-budding complexes. **a** Localization of Arrdc1 and TSG101 in the microvillus tips of HEK293T cells. Arrdc1_GFP was represented with pseudo-color coding according to fluorescence intensity. Cropped images from selected microvilli (yellow arrows) are displayed from i to v (left). Recruitment of TSG101 to the membrane in *Arrdc1*-overexpressing cells (right). **b** Co-clustering of Arrdc1_GFP and TCRβ into the cSMAC. Arrdc1_GFP[+] OTII CD4[+] T cells were stained with anti-TCRβ (H57Fab-Alexa594) and examined as described in Fig. 3a. Images were selected at 300-s post-recording, and the representative trajectories are shown in the bottom panel. **c** Release of TCR[+]/Arrdc1_GFP[+] TMPs during the synapse–kinapse transition stage. Arrdc1_GFP[+] OTII CD4[+] T cells were stained with anti-TCRβ and observed as described in Fig. 3d. **d** Purified TMPs were ultracentrifuged with a sucrose gradient and blotted with anti-Arrdc1, anti-CD3ζ, and anti-β-actin (lanes 1–6, sucrose fraction; P, pellet). **e** Knockdown of *Vps4* and *Arrdc1* by siRNA and confirmation of siRNA efficiency by western blot analysis. Surface analysis of CD4[+] T cells by SEM after knockdown of the indicated proteins. **f**, **g** Representative SEM analysis of *Vps4*-deficient OTII CD4[+] T cells on the lipid bilayer (**f**). Particle diameters and numbers of TMPs per cell were analysed by ImageJ software (**g**). Data represent the mean of three experiments ± SEM. *$P < 0.05$ vs. scrambled (Sc., left) or scrambled (20–40 nm, right) siRNA. **$P < 0.001$ vs. scrambled siRNA (40–100 nm). **h** TCR[+] TMP release on an OVA peptide/I-A[b] lipid bilayer with or without ICAM-1. The number of TCR[+] TMPs per cell was analysed by ImageJ software. Data represent the mean of three experiments ± SEM. *$P < 0.001$ vs. pMHC/ICAM-1. **i** TCR[+] TMP release on plates coated with various antibodies. Data represent the mean of three experiments ± SEM. *$P < 0.001$ vs. PLL

minimizing their influence on bystander cells. Therefore, we propose that, as opposed to exosomes and ectosomes, which are recognized as important mediators of long-distance intercellular communication, TMPs, as well as neutrophil trails and migrasomes, should be categorized as a new class of extracellular vesicles.

In this study, we found many short- or long-bridges between each T cell and surrounding antigen-loaded APCs. Indeed, these bridges might represent a structure similar to that of membrane nanotubes, which connect a wide variety of cells, including T cells, natural killer cells, B cells, and macrophages[23,39–41]. Therefore, it is necessary to revisit the process of nanotube formation. Previous reports suggest that nanotubes are formed de novo by membrane protrusions in a process driven by the actin cytoskeleton[23]. However, if nanotubes originate from microvilli, membrane nanotubes might not be generated de novo, but rather could be elongated by the "pulling force" when two synaptic cells separate. Moreover, unequivocal evidence for nanotube-mediated exchange of proteins between immune cells is still lacking.

We observed that TMPs were only transferred to the surface of physically interacting DCs and showed little effect on bystander DCs, suggesting that TMPs are primarily required for cognate DC activation. However, if TMPs are truly critical for DC activation, why do T cells need stable synapses for DC activation, as DC activation occurs within a stable IS via cognate TCR–MHC interaction along with roles involving a variety of cell–cell signals, CD40/CD40L interactions, cytokines, and adhesion molecules? A recent report demonstrated that all major subsets of T cells spend more time in kinapse mode on continuous stimulatory surfaces[22]. This is interesting, because the report suggests that activation of either T cells or DCs does not necessarily require a stable synapse if there is a strong interaction between cognate T cells and DCs. In this situation, mutual exchange of message-containing vectors between two cells could be more efficient for sustaining activation. Moreover, it has been demonstrated that T cell and DC synapses comprise several tens of submicronic contact spots and lack the large-scale concentric organization characteristic of prototypical synapses[42,43], suggesting that it is inappropriate to consider the concentric structure as a mature synapse and multifocal structures as immature[42].

Another major question concerns whether DC activation by TMPs is transduced by cognate TCR–MHC interaction or other mechanisms. First, full DC activation by 4-TMPs purified from antibody (CD3)-activated CD4[+] T cells clearly indicated that DC activation was not mediated through MHC class II molecules, whereas T cells require cognate MHC-peptide recognition for activation. In agreement with our results, many reports show that MHC class II signaling does not play a role in DC activation, whereas antigen-specific T cell interaction might contribute via other mechanisms to regulate DC activation[44–46]. Indeed, MHC

class II signaling is coupled with the induction of rapid cell death and plays opposing roles to CD40 in DC survival. However, in this regard, our results differed from those of a previous report demonstrating that B cell-specific calcium signaling triggered by TCR-enriched microvesicles is dependent upon specific interactions between TcrAND present on microvesicles and cognate moth cytochrome c-I-E[k] complexes[10]. This discrepancy might be due to the fact that DCs and B cells are activated in different ways. Indeed, several reports indicate that B cell activation requires contact-mediated signals from T cells[47], with the first of such signals identified as involving MHC class II molecules[48,49].

Enrichment of CD2 in TMPs, as well as the significant reduction of calcium responses in 4-TMP-treated DCs following use of an anti-CD2 antibody, demonstrated that CD2 plays an important role in DC activation. In agreement with this finding, CD2 the physiological trigger of human monocyte TNF and IL-1 release[29]; however, the lack of complete blockage of calcium response by the anti-CD2 antibody suggested the involvement of other factors in DC activation. Indeed, TMPs are enriched with many important T cell proteins, including membrane proteins and cytokines, among which IL-33, IL-4, IL-7, and TNF-α are well-known regulators of DC activation or differentiation[31–34]. However, because some cytokines, such as IL-1ra and IL-10, reportedly downregulate DC functions, more cautious observations are required. Furthermore, TMPs, like exosomes, can also contain genetic information, such as coding and noncoding RNAs, able to transfer T cell messages to recipient cells[50]. Further studies are underway to determine the mechanisms of TMP internalization and their control of DC activation.

In conclusion, lymphocytes are highly specialized cells that circulate throughout the entire body to scan a specific antigen and deliver messages to target cells. Because of their dynamic motility, lymphocytes might have evolved to harbor specialized mechanisms to promote communication between interacting cells, even under mobile conditions. Therefore, microvillus-derived message transfer might be a unique means of communication for lymphocytes or immune cells and not cells occupying tissues. A schematic model describing TMP generation during T cell synapse and kinapse is shown in Supplementary Fig. 9. A detailed understanding of how TMPs are exchanged between immune cells would permit therapeutic exploitation of these vesicles, resulting in their potentially serving as promising vectors for immunotherapy in diseases of the immune system and other scenarios.

## Methods

**Reagents and antibodies**. Antibodies against calcium pump pan PMCA ATPase (ab2825), Calnexin (ab22595), Vstm5 (ab179816), sodium potassium ATPase (ab58475), LAMP (ab24170), Arp3 (ab181164), CD2 (ab219411), CD40 ligand (ab52750), CD28 (ab205136), Arrdc1 (ab123600), Vps4a/b (ab229806, ab224736),

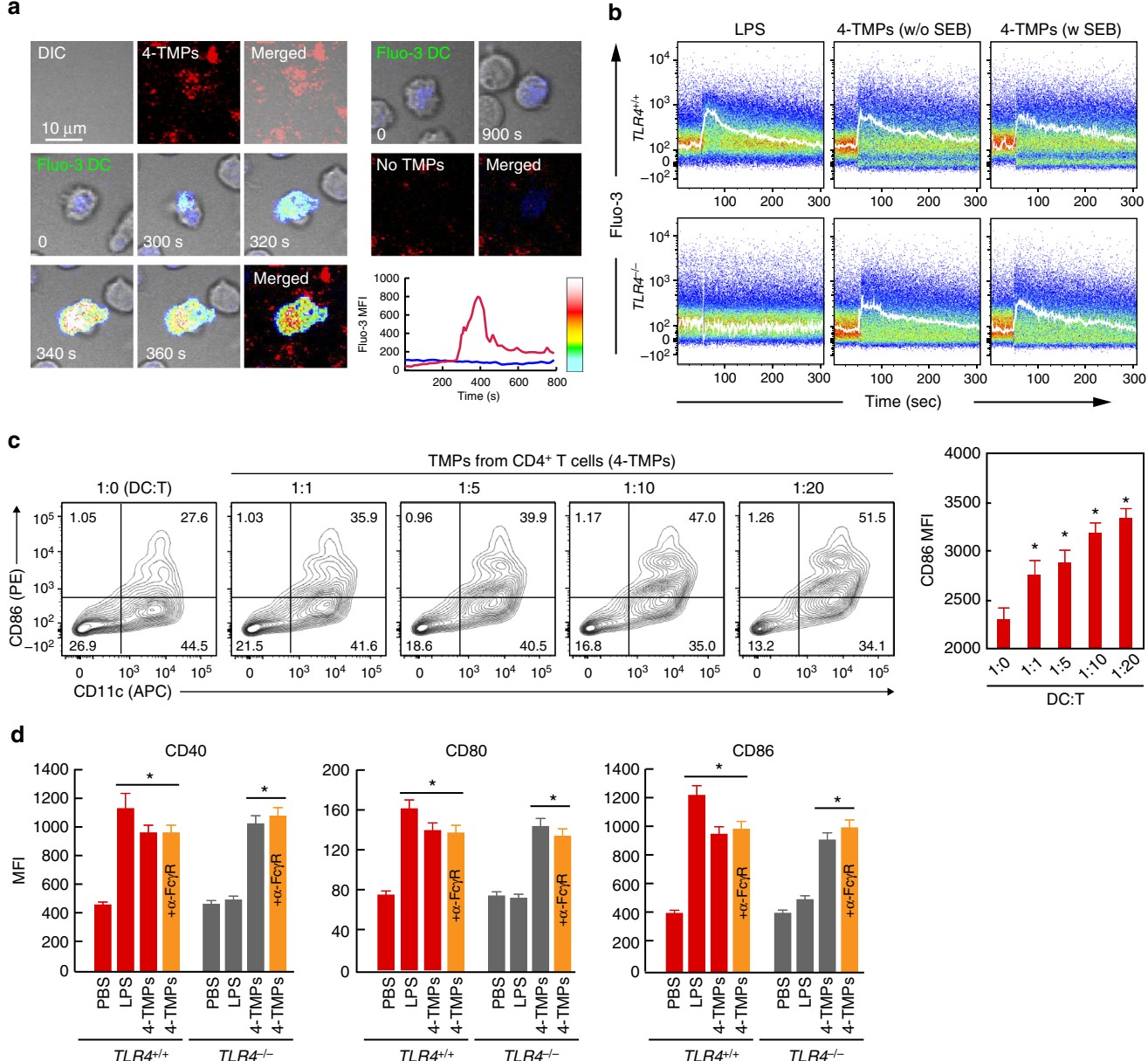

**Fig. 8** 4-TMPs activate DCs regardless of TCR engagement. **a** Representative confocal images of Fluo-3-loaded DCs placed on a 4-TMP-deposited lipid bilayer. **b** Fluo-3-loaded DCs from wild type or $TLR4^{-/-}$ mice were stimulated with 4-TMPs purified from CD4$^+$ T cells in the presence or absence of SEB. Calcium changes were analysed by flow cytometry for 300 s. Data are representative of at least three separate experiments. **c** Surface protein expression on DCs induced by 4-TMPs. DCs ($5 \times 10^6$) from wild-type mice were treated with 4-TMPs obtained from CD4$^+$ T cells ($0.05$–$1 \times 10^8$) at the indicated ratio (DC:T = 1:1 to 1:20). After 24 h, the expression of CD86 was analysed by flow cytometry and expressed as the mean fluorescence intensity (MFI). Data represent the mean of three experiments ± SEM. *$P < 0.01$ vs. 1:0. **d** DCs ($5 \times 10^6$) from wild type or $TLR4^{-/-}$ mice were treated with 4-TMPs ($5 \times 10^7$). After 24 h, the expression of CD86, CD40, and CD80 was determined as described in **c**. In some cases, FcγR was blocked 30 min before 4-TMP treatment. *$P < 0.01$ vs. PBS-treated group

and TSG101 (ab125011) were purchased from Abcam (Cambridge, MA, USA). Antibodies against TCRα (sc-9100), TCRβ (sc-73121), CD3ζ (sc-1239), CD3ε (sc-137095), CXCR4 (sc-53534), CD43 (sc-7054), and MHC class I (sc-59199) were purchased from Santa Cruz Biotechnology (Dallas, TX, USA). Antibodies against Rab5 (3547), HSP90 (4877), LCK (2752), phospho-LCK (2751), Fyn (4023), ZAP70 (2705), Syk (13198), phospho-paxillin (2541), phospho-Akt (4060), β-actin (8457), Rab11 (3539), anti-rabbit IgG (7074), and anti-mouse IgG (7076) were obtained from Cell Signaling Technology (Danvers, MA, USA). All primary antibodies for western blot were used at a dilution of 1:1000. Alexa Fluor 647-conjugated anti-human LAMP-1 (328611), allophycocyanin-conjugated anti-CD11a (141009), anti-CD44 (103011), and anti-CD2 (100111) antibodies were obtained from BioLegend (San. Diego, CA, USA). Anti-FcγR blocking antibody (553141; 1 μg per million cells in 100 μL) was purchased from BD Biosciences (San Jose, CA, USA). Anti-TCRβ (H57-597) and anti-CD154 (MR-1) were purchased from Bio-X-Cell (West

Lebanon, NH, USA). Fab preparation kit was purchased from Thermo Fisher Scientific (Waltham, MA, USA). Antibodies for fluorescein isothiocyanate-conjugated CD40 (MA5-16506), MHCII (11-5322-82), CD11a (11-0111-82), phycoerythrin-conjugated CD86 (12-0862-82), CD80 (12-0801-82), CD25 (12-0251-82), CD11c (MCD11c04), allophycocyanin-conjugated CD4 (17-0041-82), CD40L (17-1541-82), CD11c (17-0114-82), and PercpCy5.5-conjugated CD28 (45-0281-80) were purchased from eBioscience (San Diego, CA, USA). All antibodies for flow cytometry were used at a dilution of 1:100. Proteome Profiler Mouse Cytokine XL array kit and mouse-TNFα, -IL-4, and -IL-7, and -IL-33 cytokine DuoSet enzyme-linked immunosorbent assay (ELISA) kits were purchased from R&D Systems (Minneapolis, MN, USA). Avidin, tetra-methylrhodamine (TRITC)-phalloidin, poly-L-lysine (PLL), anti-rabbit IgG-TRITC, A23187, and phorbol 12-myristate 13-acetate (PMA), latrunculin A (Lat A), jasplakinolide (JAS), and CK636 were purchased from Sigma-Aldrich (St.

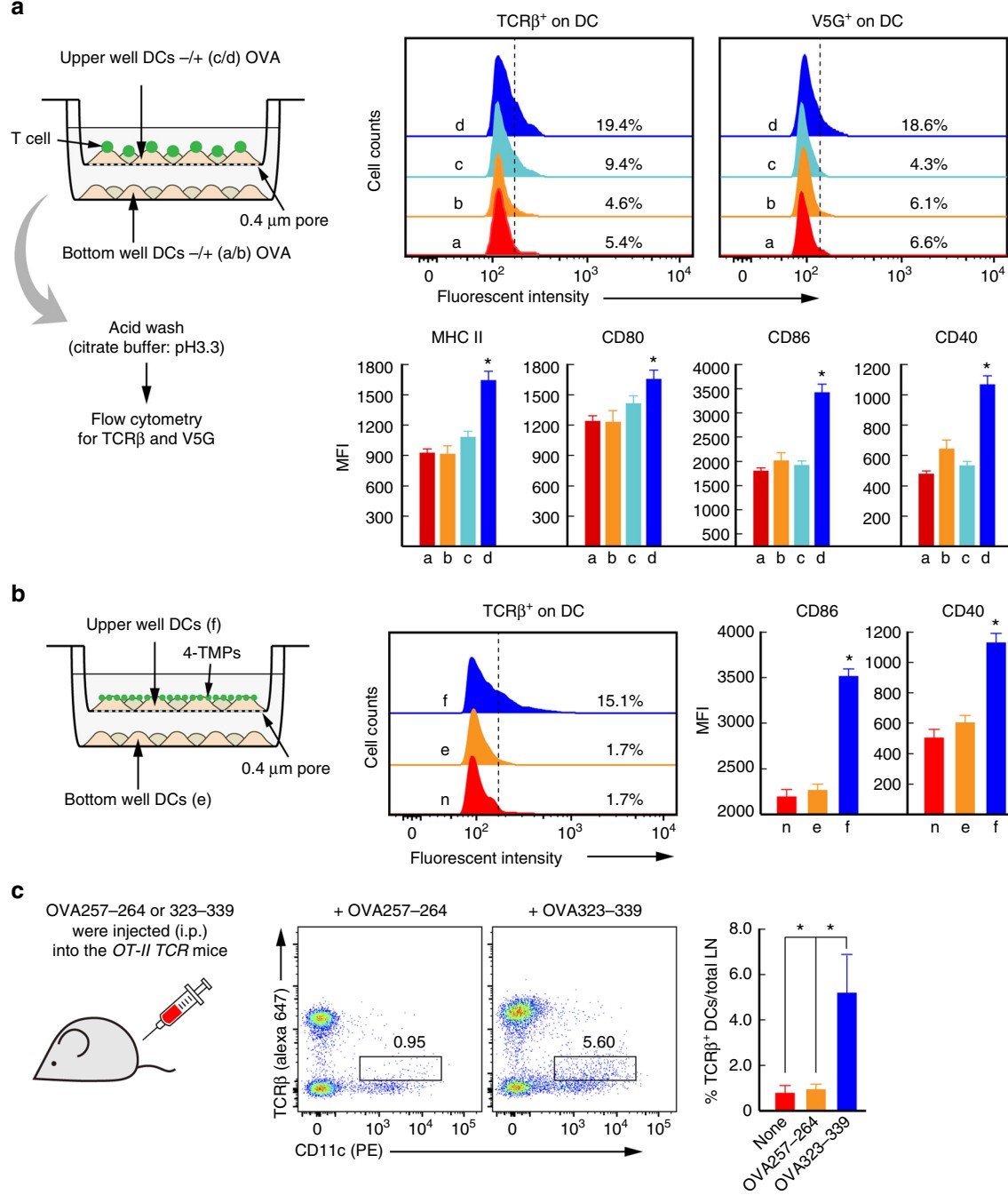

**Fig. 9** Physical interaction between TMPs and the DC surface is critical for TMP-mediated DC activation. **a** Schematic diagram of the transwell system (left). After co-incubation for 2 h, DCs were acid washed and evaluated for TCRβ and V5G transfer (upper panel), followed continued incubation for 24 h. The expression of MHC class II, CD80, CD86, and CD40 on DCs (CD11c[+]) was analysed by flow cytometry and expressed as histograms and mean fluorescence intensity (MFI). Data represent the mean of three experiments ± SEM. *$P < 0.01$ vs. **c**. **b** 4-TMPs from anti-CD3-coated surfaces were treated with DCs as depicted in the schematic diagram (left) and incubated as described in **a**. TCRβ, CD86, and CD40 presentation on DCs (CD11c[+]) was analysed as described in **a**. Data represent the mean of three experiments ± SEM. *$P < 0.01$ vs. **c**. **c** Transfer of TCRs to DCs in vivo. *OTII TCR* transgenic mice were administered OVA257-264 and OVA323-339 (100 μg; i.p.) in PBS, respectively. After 48 h, draining lymph nodes were taken and dissociated, and cells were treated by acid buffer and stained for the indicated surface markers. Boxed areas represent TCRβ expression on DCs. Data are represented by one of three independent experiments with similar results. *$P < 0.01$ vs. none or OVA257-264

Louis, MO, USA). Alexa594-labeling kit, Lipofectamine 2000, Fluo-3/AM, Cell-Tracker CMFDA-green, CMRA-Orange, and Deep Red dyes were purchased from Invitrogen (Carlsbad, CA, USA). Hybridoma cell lines for mouse anti-CD3 (145-2C11; CRL-1975), mouse anti-CD28 (PV1; HB-12352), and anti-human CD3 (OKT3; CRL-8001) were purchased from the American Type Culture Collection (Manassas, VA, USA). Anti-human ICAM-1-His stable S2 cell line was a gift from Dr. M.L. Dustin (The Kennedy Institute of Rheumatology, Oxford, UK).

Staphylococcal enterotoxin E (SEE) and SEB were purchased from Toxin Technology, Inc. (Sarasota, FL, USA). OVA peptide fragments (323–339 and 257–264) were purchased from GeneScript (San Francisco, CA, USA). Reverse transcription PCR premix and restriction enzyme were purchased from Enzynomics (Daejeon, Korea). Plasmid DNA purification kit and WEST-ZOL western blot detection kit were purchased from Intron Biotechnology (Seongnam, Korea). PrimeSTAR HS DNA polymerase was purchased from TaKaRa Bio (Shiga, Japan). Endo H and

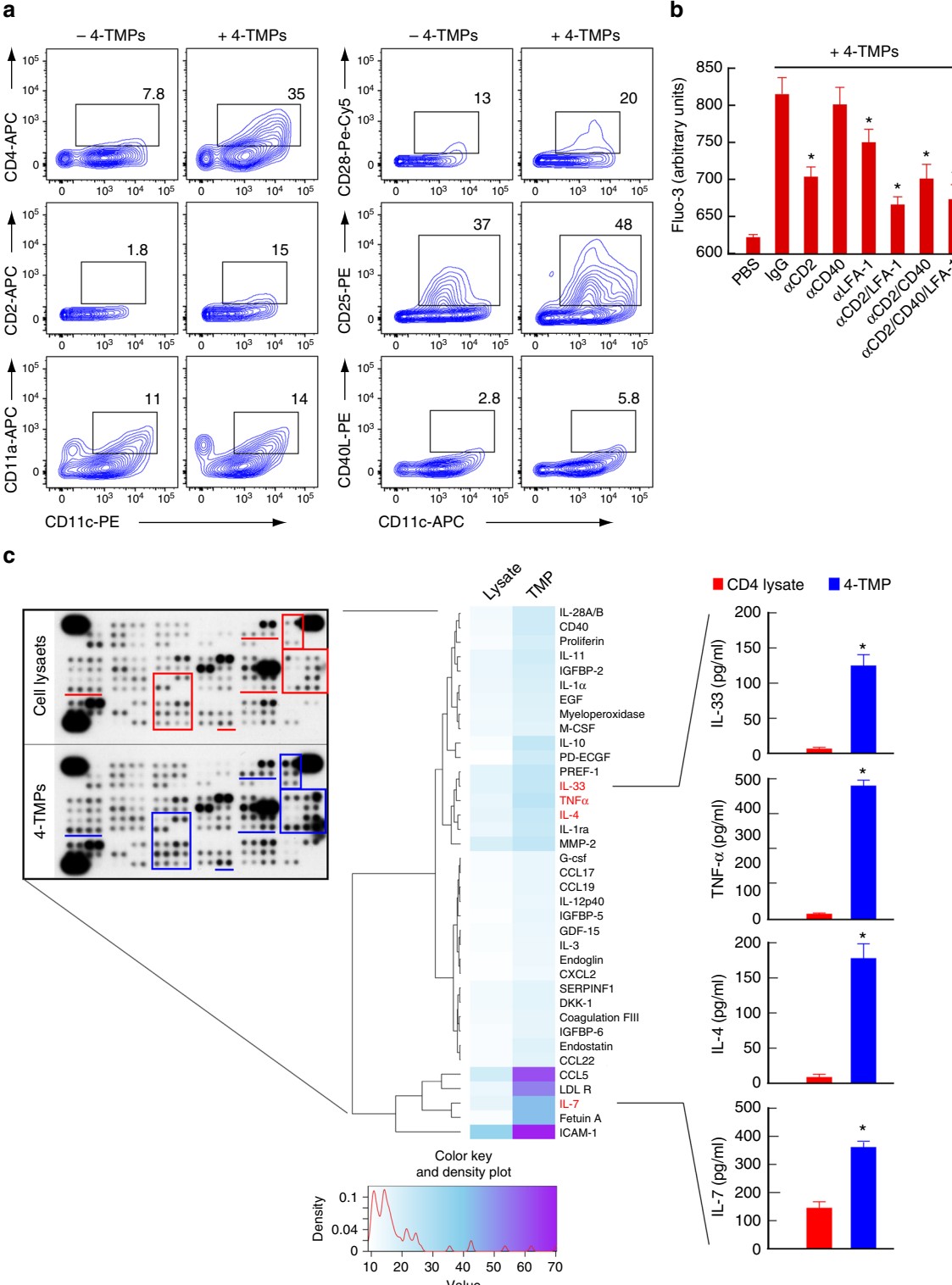

**Fig. 10** CD2/CD58 and LFA-1/ICAM-1 play important roles in TMP-mediated DC activation. **a** DCs (5 × 10⁶) were treated with either PBS or 4-TMPs (5 × 10⁷) for 2 h, and the transfer of indicated proteins on DCs (CD11c⁺) was analysed by flow cytometry. Data represent the mean of three experiments ± SEM. *P < 0.01 vs. no 4-TMPs. **b** Fluo-3-loaded DCs (5 × 10⁶) were pretreated for 30 min with the indicated blocking antibodies (anti-CD2, -CD40, and -LFA-1), and then stimulated with 4-TMPs (5 × 10⁷). Calcium signals were expressed as arbitrary units (a.u.) of Fluo-3 fluorescence from the average of at least three separate experiments. Data represent the mean of three experiments ± SEM. *P < 0.01 vs. PBS-treated group. **c** Representative images of cytokine arrays (left) and the heatmap showing cytokine-expression levels in total CD4⁺ T cell lysate and 4-TMPs (right). Each sample (200 μg) was hybridized to membranes containing capture antibodies specific for 112 cytokines. Data were quantified by densitometric analysis and statistically analysed. Cytokines overexpressed by up to 1.5-fold in TMPs relative to their expression in lysate are highlighted by red and blue boxes. Some cytokines were further confirmed by ELISA. Data represent the mean of three experiments ± SEM. *P < 0.01 vs. CD4⁺ T cell lysate

PNGase F were purchased from New England Biolabs (Ipswich, MA, USA). Lentiviral vector for LifeAct_mRFP was purchased from Ibidi (Verona, WI, USA). *TSG101, Vps4a-, Vps4b-,* and *Arrdc1-*targeting and scrambled control small-interfering (si)RNAs were purchased as a pool of four siRNA duplexes from Dharmacon (Lafayette, CO, USA). Blocking peptides for anti-CD2 and -CD3ε were purchased from MyBioSource (San Diego, CA, USA). MHC I-A^b was provided by the NIH Tetramer Core Facility (Atlanta, GA, USA). Amaxa Nucleofector II and the Mouse T Cell Nucleofector kits were purchased from Lonza (Walkersville, MD, USA). Complete protease inhibitors and phosphatase inhibitors were purchased from Roche Applied Science (Indianapolis, IN, USA). Transwell apparatus (0.4 μm) were purchased from Costar (Corning, NY, USA).

**Cells**. Jurkat T cells (TIB-152), HEK293T (CRL-1573), Raji B cells (CCL-86; all from ATCC), and Platinum-E (Plat-E) retroviral packaging cells (Cell Biolabs, San Diego, CA, USA) were maintained in RPMI-1640 or Dulbecco's modified Eagle medium (Invitrogen) supplemented with 10% (v/v) fetal bovine serum (FBS; Invitrogen). Naïve CD3+ T cells were purified from the mouse spleen and lymph nodes by negative selection using a T cell enrichment column (R&D Systems). To generate mouse T cell blasts, CD3+ T cells were incubated in 2 μg/mL anti-CD3/28-coated culture plates with 100 U/mL rIL-2 for 48 h and cultured for a further 5 days with 100 U/mL rIL-2. Mouse splenocytes were dispersed and purified into CD4+, CD8+, and CD19+ populations using the EasySep magnetic separation system (Stemcell Technologies, Vancouver, Canada) or MACS cell separation (Miltenyi Biotec, Bergisch Gladbach, Germany). The purity of each population was confirmed as >95% by flow cytometry. To establish B cell blasts, CD19+ cells from C57BL/6 wild-type mice were activated with LPS (10 μg/mL) for three days in complete RPMI. Bone marrow was flushed from femur and tibia bones, and bone-marrow-derived DCs (BMDCs) were grown with the addition of 20 ng/mL granulocyte macrophage-colony stimulating factor for five days.

**Animals**. C57BL/6 wild-type mice and *OTII TCR* transgenic mice (C57BL/6 background) were purchased from the Jackson Laboratory (Bar Harbor, ME, USA). *Vstm5*-knockout mice were generated by Macrogen using CRISPR/Cas9 (Seoul, Korea). *LFA-1*-knockout mice were provided by Dr. Minsoo Kim (University of Rochester, NY, USA). *TLR4*-knockout mice were provided by Dr. Sang-Myeong Lee (Chonbuk National University, Iksan, Korea). All mice were housed in specific pathogen-free conditions. All experimental methods and protocols were approved by the Institutional Animal Care and Use Committee of the School of Life Sciences, Gwangju Institute of Science and Technology (Gwangju, Korea) and performed in accordance with their approved guidelines.

**cDNA constructs**. The coding sequences for human or mouse CD3ζ, Vstm5, TSPAN25, TSG101, and Arrdc1 were cloned by PCR using human or mouse CD3+ T cell cDNA as a template and verified by sequencing (Macrogen). Vstm5ΔN4 (ΔCNQC) was constructed from wild-type Vstm5 cDNA by mutagenesis. All obtained PCR products were inserted into the pEGFP_N1, ptdTomato_N1, or pHJ-1 (CMV promoter) lentiviral vector. PCR was performed with the following primers (the respective forward and reverse pairs are indicated): human CD3ζ, 5′-TTGCTAGCGATGAAGTGGAAAGGCGCTTTTCACC-3′ and 5′-TTTCTCGAGGCGAGGGGGCAGGGCCTGCATGTG-3′; mouse CD3ζ, 5′-TTTTTA GATCTCATGAAGTGGAAAGTGTCTGTTCTCGCCTGC-3′ and 5′-TTTCGCGGCCGCGCGAGGGGCCAGGGTCTGCATATGCAGGGC-3′; human Vstm5, 5′-TTGCTAGCGATG AGGCCTCTGCCCAGCGGGAGGAGGAAG-3′ and 5′-TTTGAGATCTGAGCCTCCGGC AGCACACTCAACATCTTCCAGCTCAATCTC-3′; mouse Vstm5, 5′-TTTTTAGATCTC ATGAGACCACTGCGGGTGCGGGGAGAGGACC-3′ and 5′-TTTCAGAATTCGGCCTCC GGCAGCACACTCAACTTCTTTCA TCTCAATCTC-3′; human TSPAN25, 5′-AAGCTA GCGATGGGCATGAGTAGCTTGAAACTGCTGAAG-3′ and 5′-TTTCAGAATTCGGCC TCCGGCAGCGCCTCCGGCAGCTAGCCCTATGGTCTGGCTGGTTTTTGTCAAT-3′; mouse TSPAN25, 5′-AAGCTAGCGATGGGCATGAGCAGCCTGAAATTGCTGAAA-3′ and 5′-TTTCAGAATTCGGCCTCCGGCAGCTCCGGCAGCCAGCCCTAA AGCCTG GCTTGTTTTGTCAAT-3′; mouse TSG101, 5′-GCAAGGAAGACTGCGGGCCTTAGT GACCTCTACACGGTACCGAAA-3′ and 5′-AAACGGTAC CGTGTAGAGGTCACTAAG GCCCGCAGTCTTCCTTGC-3′; and mouse Arrdc1, 5′-AAATCAGATCTCATGGGGAGGGTG CAGCTCTTCGAGATCCG C-3′ and 5′-AAACAGAATTCGGCTTCCTGGGATCAGGCC AAGGTCTGTA CC-3′.

**Reverse transcription PCR and real-time quantitative (q) PCR**. Total RNA was isolated from cells or homogenized tissues of C57BL/6 mice with TRI reagent (Molecular Research Center, Cincinnati, OH, USA) and reverse transcribed using RT-Premix (Intron Biotechnology). PCR was performed with the following primers (the respective forward and reverse pairs are indicated): mouse *Vstm5*, 5′- AATG TCTCCCAAAGCCACAG-3′ and 5′- TAGCGGATCTCAGAC ACACG -3′; mouse *TSPAN25*, 5′-GACCATAGCCATCCTGCTCT-3′ and 5′-TGGATGAAGT CCCATGCCTT-3′; and mouse glyceraldehyde 3-phosphate dehydrogenase (*Gapdh*), 5′-GCACAGTCAAGGCCGAGAAT-3′ and 5′-GCCTTCTCCATGGT GGTGAA-3′. The expression levels of *Vstm5* and *TSPAN25* were evaluated by qPCR. Amplification was performed in a StepOne real-time PCR system (Applied

Biosystems, Norwalk, CT, USA) for continuous fluorescence detection in a total volume of 10 μL of cDNA/control and gene-specific primers using SYBR Premix Ex Taq (TaKaRa Bio). The mRNA levels of the target genes were normalized relative to those of *Gapdh* using the following formula: relative mRNA expression = $2^{-(\Delta Ct \text{ of target gene} - \Delta Ct \text{ of GAPDH})}$, where Ct is the threshold cycle value. In each sample, the expression of the analysed gene was normalized to that of *Gapdh* and described as the mRNA level relative to *Gapdh*.

**Cell transfection and viral infection**. To establish stable cell lines, cDNAs in the pHJ-1 lentiviral vector were co-transfected with lentiviral-packaging vectors (pHDM-Hgpm2, pRC/CMV-Rev1b, and pHDM.G) into 293 T cells. The supernatants were then collected and spin-infected into Jurkat T cells by centrifugation at 2000 × g for 90 min at 25 °C in the presence of 8 μg/mL polybrene (Sigma-Aldrich). For retroviral transduction, mouse CD4+ T cells from C57BL/6 or *OTII TCR* transgenic mice were incubated in 2 μg/mL anti-CD3/28-coated culture plates with 100 U/mL rIL-2 for 48 h, and 1 × 10^6 retroviral packaging cells (platinum-E cells) were plated overnight in 6-cm2 dishes. Retroviral particles were generated by transfection with V5G_pMSCV and the pCL-Eco packaging vector using Lipofectamine 2000 (Invitrogen). After 48 h, viral supernatants (2 mL) were harvested, mixed with 1 × 10^6 mouse T cells, placed on 20 μg/mL RetroNectin (Clontech, Mountain View, CA, USA)-coated 12-well plates, and centrifuged at 2000 × g for 90 min at 25 °C with rIL-2 (100 U/mL). The transduced T cells were maintained with fresh mouse T media with rIL-2 and expanded for five days. Transient transfection of mouse T cells and HEK293T cells for gene expression or knockdown was performed with an Amaxa Nucleofector II and the Mouse T Cell Nucleofector Kit (Lonza) and Lipofectamine 2000 (Invitrogen), respectively.

**Western blot analysis**. Cells and isolated TMPs were lysed in ice-cold lysis buffer (50 mM Tris-HCl (pH 7.4), 150 mM NaCl, 1% Triton X-100, 1 × complete protease/phosphatase inhibitor cocktail) for 1 h on ice. Lysates were centrifuged at 16,000 × g for 25 min at 4 °C, and the harvested supernatants were mixed with sodium dodecyl sulphate (SDS) sample buffer (100 mM Tris-HCl (pH 6.8), 4% SDS, 20% glycerol, and bromophenol blue) and then heated for 5 min. Proteins were separated by 10% to 12% SDS polyacrylamide gel electrophoresis (PAGE) and transferred to nitrocellulose membranes using a Trans-Blot SD semi-dry transfer cell (Bio-Rad, Hercules, CA, USA). Membranes were then blocked in 5% skim milk for 1 h, rinsed, and incubated with primary antibodies in Tris-buffered saline containing 0.1% Tween 20 (TBS-T) and 3% milk overnight. Excess primary antibody was removed by washing the membrane four times in TBS-T before incubation with peroxidase-labeled secondary antibody (0.1 μg/mL) for 1.5 h. Bands were visualized with a WEST-ZOL western blot detection kit and exposed to X-ray film. For determination of Vstm5 localization in the membrane, V5G+ Jurkat T cells were lysed, and 10 μg of protein of the total lysate was treated with *N*-glycosidase F (PNGase F) and endoglycosidase H (EndoH) (both from New England Biolabs) according to manufacturer instructions and subjected to western blot. All uncropped blots in the main figures are shown in Supplementary Figs. 10 through 12.

**Subcellular fractionation**. Plasma membrane, cytosol, and vesicle fractions were obtained by differential centrifugation[51,52]. Cells were washed, mixed with ice-cold hypotonic buffer (300 mM sucrose, 1 mM EDTA, 20 mM Tris-HCl (pH 7.4), and complete protease inhibitors), and mechanically lysed with a Dounce homogenizer (40 strokes). Samples were first spun at 4000 × g at 4 °C for 10 min to remove debris and nuclei, and the supernatants were centrifuged at 17,000 × g for 30 min at 4 °C for the membrane fraction (pellet). Supernatants were recovered and transferred into clean ultra-microcentrifuge tubes and centrifuged at 100,000× g for 60 min at 4 °C for vesicle (pellet) and cytosol fractions (supernatant). Endoplasmic reticulum (ER)/mitochondria fractions were obtained as previously described[53]. Briefly, 2.5 × ER/mitochondria stabilization buffer (525 mM mannitol, 175 mM sucrose, 2.5 mM EDTA (pH 7.5), and 12.5 mM Tris-HCl (pH 7.5)) was added to samples mechanically lysed with a Dounce homogenizer (40 strokes), as described. After centrifugation at 1300× g for 5 min at 4 °C, the supernatant was recovered and transferred to clean microcentrifuge tubes. The first pellet sample contained the nuclei, intact cells, and cell debris. Supernatants were centrifuged at 17,000× g for 15 min at 4 °C for the ER fractions.

**Preparation of supported planar lipid bilayers**. Bilayers were prepared as described previously[54,55], where lipids (DOPC, Biotin CAP-PE, DOPE-PEG5000, and DOGS-NTA nickel salt; Avanti Polar Lipids, Birmingham, AL, USA) were reconstituted in chloroform at 95.5:1:0.5:3 mol%, respectively. The mixture was then dried under nitrogen and desiccated in a vacuum chamber overnight. The dried lipid cake was hydrated in PBS, sonicated using a tabletop sonicator for 15 min to generate multilamellar vesicles, and passed through a 50-nm pore membrane using a mini-extruder (Avanti Polar Lipids). Glass slides (25 × 75 mm; #1.5; Thermo Fisher Scientific) were cleaned for 15 min using acetone, 10 N NaOH for 10 min, and 1 M HCl for 10 min and washed thoroughly with MilliQ water. Slides were then dried and placed in an air plasma cleaner (Harrick Plasma, Ithaca, NY, USA). To create a bilayer within the flow cell, a 2-μL liposome drop was deposited on the glass slide between strips of double-sided tape (Ibidi Sticky-Slide

VI0.4 Luer closed chambers; Ibidi, Martinsried, Germany), after which glass coverslip was placed on top of the glass slide across the double-sided tape, simultaneously allowing a single planar bilayer to form on the coverslip surface and creating a flow chamber. After thorough rinsing with PBS, the chambers were incubated with ICAM-1 6× His (250–300 molecules/μm2), followed by avidin (1 μg/mL), and then thoroughly rinsed again with PBS and incubated with monobiotinylated MHC I-A$^b$ (20–50 molecules/μm2) for 15 min[56]. Densities of molecules on the bilayer were measured as previously described[55]. Chambers were rinsed and left in HEPES-buffered saline (HBS) supplemented with 1% human serum albumin. Lipid bilayers were used for imaging studies on the same day.

**TIRFM**. To monitor the centripetal movements of TCRβ, V5G, or Arrdc1_GFP, V5G$^+$ or kinapse, Arrdc1_GFP$^+$ OTII CD4$^+$ T blasts were stained with TCRβ (H57Fab-594). The cells were then placed on a planar lipid bilayer presenting OVA323-339-I-A$^b$/ICAM-1 and immediately imaged for 10 min to 30 min by TIRFM (IX-81; Olympus, Tokyo, Japan) equipped with a solid-state laser (488 nm, 20 mW; Coherent, Santa Clara, CA, USA). In some cases, cells were pretreated with actin modulators, such as Lat A (237 nM), JAS (100 nM), and CK636 (100 μM), for 30 min before imaging. To examine the generation of V5G$^+$, TCRβ$^+$, or Arrdc1_GFP$^+$ microvillus particles during T cell synapse or kinapse, OTII CD4$^+$ T blasts as described were imaged for long periods (~2 h) with TIRFM.

**Confocal microscopy**. Jurkat T, mouse CD3$^+$ T, CD4$^+$ T, OTII CD4$^+$ T, or HEK293T cells expressing V5G; various tetraspanin proteins including TS25G, CD3 ζ_ptdTomato, Arrdc1_GFP, or _RFP; or TSG101_GFP in HEPES imaging buffer were placed on PLL- or anti-CD3/28-coated coverglasses. For LAMP-1 imaging, cells were fixed with 4% paraformaldehyde. Cells were then permeabilised with 0.1% Triton X-100 in PBS, blocked with 1% BSA/PBS for 1 h, rinsed with PBS, and incubated with anti-LAMP-1 antibody overnight at 4 °C. Secondary antibodies were added after washing and incubated in the dark for 1 h at room temperature. For actin staining, cells were incubated with TRITC-phalloidin in PBS for 30 min at room temperature. To visualize V5G$^+$ microvilli and actin under live conditions, V5G$^+$ Jurkat T cells were transfected with LifeA_RFP and then incubated for the indicated durations (5–30 min) with 1 μg/mL SEE-pulsed Raji B cells. The cells were then imaged using a 100× NA 1.40 oil immersion objective and an FV1000 laser scanning confocal microscope (Olympus).

To monitor TMP release under live conditions, DCs (5 × 10⁴) on glass-bottomed dishes were stained with Cell Tracker Deep Red Dye (1 μM) for 30 min and washed with warm PBS. V5G$^+$ CD4$^+$ T cells (5 × 10⁵) were then placed on DCs and incubated in the presence of SEB (5 μg/mL). T cell movement and V5G$^+$ MP release were monitored using a W Plan-Apochromat 40 × /1.0 objective and a Zeiss LSM 880 confocal laser scanning microscope (Carl Zeiss, Oberkochen, Germany). V5G$^+$ MP generation was also determined from activated T cells after incubation with mouse B cells or DCs in the presence or absence of 5 μg/mL OVA323-339 for 30 min. To quantitate the V5G$^+$ MPs generated during T cell synapse or kinapse, mouse V5G$^+$ OTII CD4$^+$ T blasts (5 × 10⁵) were incubated on BMDCs for 1 h, and V5G$^+$ MPs were quantified as the number within a 50 μm2 × 50 μm2 area around a single cell using Imaris software (Imaris, Zurich, Switzerland).

To quantitate TCRβ$^+$ MPs, OTII CD4$^+$ T or CD4$^+$ T blasts (5 × 10⁵) were stained with TCRβ (H57Fab-594), and the cells were placed on a planar lipid bilayer presenting OVA323-339-I-A$^b$ with/without ICAM-1 or coverglasses coated with various antibodies against CD3, LFA1 (CD11a), MHC class I, CD43, and CD44 for 1 h at 37 °C. At least 20 images for each group were obtained using a 100× NA 1.40 oil immersion objective and an FV1000 laser scanning confocal microscope (Olympus).

**TMP quantitation**. OTII CD4$^+$ T blasts (5 × 10⁵) were placed on anti-CD3 antibody lipid bilayers presenting OVA peptide/I-A$^b$ and ICAM-1 or lipid bilayers presenting ICAM-1, or the cells were treated with anti-CD3 antibody, PMA/A23187, or ICAM-1 for 1 h at 37 °C. After incubation, the cells were fixed for SEM analysis. TMPs generated from around a single cell were quantitated as described in a proceeding subsection.

**Image analysis**. All imaging data, including the generation of TMPs (by SEM analysis) and V5G$^+$MPs and TCR$^+$MPs (by confocal analysis), around one single cell were analysed by ImageJ software (National Institutes of Health, Bethesda, MD, USA). Six major steps were applied as follows: (1) background subtraction, (2) sharpening and finding edges, (3) smoothing and converting to black and white, (4) closing and filling holes, (5) denoising and segmenting, and (6) filtering and measuring. IMARIS software (BitPlane, Belfast, Northern Ireland) was used for counting and evaluating fluorescent clusters. Cluster movements were tracked automatically using an algorithm for Brownian movement, and the maximum gap distance in one frame was set at 5 μm.

**Palmitoylation analysis**. For [$^3$H]-palmitate labeling, GFP or V5G$^+$ Jurkat T cells (2 × 10⁷) were stimulated with anti-CD3/28 for 24 h and cultured with 1 mCi [$^3$H]-palmitate for 4 h. Cells were then washed three times with PBS and lysed in RIPA buffer (10 mM Tris, 150 mM NaCl, 1% Triton X-100, 0.1% SDS, and 0.1%

deoxycholate (pH 7.35)) in the presence of protease inhibitors. After centrifugation at 16,000 × g for 25 min at 4 °C, the supernatant was incubated with 10 μg rabbit anti-GFP antibody for 4 h at 4 °C. ProteinA–Sepharose beads (25 μL; GE Healthcare, Pittsburgh, PA, USA) were added to each sample, and incubation continued for 2 h at 4 °C. Immunoprecipitates were centrifuged at 16,000 × g for 10 min and, after removal of the supernatant, washed twice in 0.5 mL ice-cold RIPA buffer. Samples were separated by SDS-PAGE, and the gels were treated with Enhance solution (NAMP 100; GE Healthcare) to improve the radioactive signal. Gels were dried in a Bio-Rad vacuum apparatus (Bio-Rad) and exposed to photographic film for at least 2 weeks at −80 °C.

**TMP isolation and purification**. To isolate TMPs, we developed a purification strategy. Mouse naive CD3$^+$ cells for TMPs or CD4$^+$ cells for 4-TMP T cells were prepared as described, where 5 × 10⁷ cells were resuspended in serum-free RPMI-1640 and incubated on anti-CD3-coated plates for 2 h at 37 °C. Cells were then removed by shaking with cold PBS, and TMPs on culture dishes were dissociated by the addition of excess anti-CD3 blocking peptides (antibody:peptide molar ratio = 1:20) in the presence of cold PBS for 30 min. All supernatants were subjected to two successive centrifugations at 2000 × g for 10 min and 2500 × g for 10 min in order to completely eliminate cells and debris and dialyzed in PBS for 24 h at 4 °C. TMPs in PBS were centrifuged for 1 h at 100,000 × g, and pellets were resuspended with PBS and subjected to western blot, SEM, TEM, and LC-MS/MS analyses and DC activation. Alternatively, purified TMPs were further ultra-centrifuged with a sucrose gradient, as described previously[27]. Briefly, TMPs were resuspended in PBS and reloaded onto a sucrose gradient of 12 different sucrose concentrations from top to bottom (10–90% sucrose) and centrifuged at 100,000 × g for 16 h. Fractions were then carefully collected at 2 mL each from the bottom of the tube. All fractionated samples were diluted with PBS and subjected to centrifugation at 100,000 × g for 90 min to pellet the vesicles.

**Sample preparation for LC-MS/MS**. The same amount of proteins from the total lysate, membrane, TMPs, and vesicles was subjected to SDS-PAGE on a 12% polyacrylamide gel and stained with Coomassie Brilliant Blue, after which protein bands in each lane were excised. After reduction with DTT and alkylation with iodoacetamide, each band was treated with trypsin to digest the proteins, and washed with 10 mM ammonium bicarbonate and 50% acetonitrile (ACN), swollen in digestion buffer containing 50 mM ammonium bicarbonate and 1 μg trypsin, and incubated at 37 °C for 16 h. Peptides were recovered by two cycles of extraction with 50 mM ammonium bicarbonate and 100% ACN. The resulting peptide extracts for each band were pooled, lyophilized, and stored at −20 °C.

**NanoLC-electrospray ionization (ESI)-MS/MS for identification using Orbitrap**. Each sample was analysed on an EASY-nLC 1200 nanoLC system equipped with a trapping column (PepMapTM100; C18 column: 75 μm × 2 cm, 3 μm; Thermo Fisher Scientific) and an analytical column (PepMapRSLC; C18 column: 75 μm × 15 cm, 3 μm; Thermo Fisher Scientific) coupled to an Orbitrap Fusion Lumos Tribrid mass spectrometer (Thermo Fisher Scientific) equipped with an EASY-Spray nanoES-ion source (Thermo Fisher Scientific). For peptide separation, the mobile phases comprised 100% water (A) and 80% ACN (B), with each containing 0.1% formic acid. The LC gradient began with 2% mobile phase B, which was linearly ramped to 6% for 1 min, 10% for 16 min, 50% for 74 min, and 100% for 1 min, with 100% maintained for 8 min before decreasing to 2% for another 5 min. The column was re-equilibrated with 2% B for 15 min before the next run. The voltage applied to produce the ES was 2.0 kV. During chromatographic separation, the Orbitrap Fusion Lumos mass spectrometer was operated in data-dependent mode, and MS data were acquired using the following parameters: full scans were acquired in the Orbitrap at a resolution of 120,000 ($m/z$ 400–1600) for each sample; data-dependent collision-induced dissociation (CID) MS/MS scans per full scan were acquired during a 3-s cycle time; CID scans were acquired in a linear trap quadrupole with 35% normalized collision energy; and a 1.2-Da isolation window for MS/MS fragmentation was applied. Previously fragmented ions were excluded for 12 s.

**Protein identification**. MS/MS spectra were searched against the target-decoy mouse Refseq database (Uniprot mouse database: 20160306 release) in IP2 (Integrated Proteomics Pipeline). The precursor mass tolerance was confined within 25 p.p.m., with a fragment-mass tolerance of 0.6 Da and an unlimited number of missed cleavages allowed. Trypsin was selected as the enzyme, carbamidomethylation at cysteine was chosen as a static modification, and oxidation at methionine was chosen as the variable modification. The output data files were filtered and sorted to construct the protein list using DTASelect, with two or more peptide assignments required for protein identification. Assigned peptides were filtered with a 5% false discovery rate. The peptide spectral matches (PSMs) of each protein were normalized by total sample PSMs in order to produce more accurate results and remove inter-experimental variation. Approximately 300 proteins that were repeatedly found in each experiment were selected and further analysed for their differential PSM among different sample groups. Unsupervised hierarchical clustering analysis and PCA[57,58] were applied to identify similarity and differences in the whole proteome among the four groups.

**Measurement of surface proteins on DCs**. BMDCs were cultured for an additional 24 h in media alone or with 100 ng/mL LPS or 4-TMPs derived from activated CD4$^+$ T cells from wild-type mice. In some cases, to rule out the influence of the anti-CD3 antibody on ligation to FcγR on DCs, an FcγR blocking antibody (clone 2.4G2; BD Pharmingen, San Jose, CA, USA) was pre-incubated with TMPs for 30 min before treatment. Cells were washed with 1% FBS in PBS and double-stained with anti-CD11C, anti-CD80, anti-CD86, and anti-CD40 monoclonal antibodies. Fluorescence intensities were measured in the resuspended cells by flow cytometry (Beckton Dickinson, Mountain View, CA, USA) and analysed using FlowJo software (TreeStar, Ashland, OR, USA).

**Measurement of calcium influx in DCs**. For live imaging of calcium influx in DCs, BMDCs were stained with 5 μM Fluo-3/AM in FBS-free RPMI-1640 for 30 min at 37 °C, washed, and incubated in complete RPMI-1640 for another 30 min. Cells were resuspended in warmed HBS/human serum albumin (HSA) buffer and introduced onto the TMP-containing coverglass. To generate TMPs on lipid bilayers, anti-TCRβ (H57Fab-Alexa594)-stained OTII CD4$^+$ T blasts were placed on the lipid bilayers presenting pMHC and ICAM-1 for 60 min, and after removing all of the T cells with cold HBS/HSA, Fluo-3-loaded DCs were introduced onto the TMP-containing coverglass and imaged after 10 min of interaction with TMPs. To quantify Fluo-3 fluorescence intensity in DCs, cells placed on "+TMPs" or "−TMPs" were randomly selected and expressed as relative fluorescence units (RFU = max−min). Calcium influx was alternatively measured by flow cytometry. BMDCs ($5 \times 10^6$) stained with 5 μM Fluo-3/AM were mixed with TMPs isolated from CD4$^+$ T blasts ($5 \times 10^7$; ratio, 1:10) and measured for 5 min. In some inhibition studies, isolated TMPs were pre-incubated with 10 μg of blocking antibodies, as indicated, for 30 min at room temperature.

**Cytokine arrays**. Mouse cytokines in the total CD4$^+$ T blasts lysate or 4-TMPs were measured using the Proteome Profiler mouse cytokine XL array kit (R&D Systems) according to manufacturer instruction. The kit consists of a nitrocellulose membrane containing 112 different anti-cytokine/chemokine antibodies spotted in duplicate. Briefly, the membranes were blocked, incubated overnight with each sample (200 μg), pre-incubated for 1 h with a biotinylated detection-antibody cocktail, washed, and detected using a horseradish peroxidase-conjugated streptavidin system. The data were presented as heatmaps, with cytokines in TMPs being 1.5-fold more than those in the CD4$^+$ T blast lysates. The lysates of total CD4$^+$ T blasts or 4-TMPs (10 μg) were also subjected to ELISA assay (R&D Systems).

**TMP transfer in vitro and in vivo**. DCs were cultured on the upper and bottom compartments of a 0.4-μm pore-sized transwell. OTII CD4$^+$ T cells or V5G$^+$ OTII CD4$^+$ T cells ($1 \times 10^6$) were added to upper-well DCs ($5 \times 10^5$) in the presence or absence of OVA323-339, and after 2 h, cells were treated with citrate buffer (0.133 M citric acid and 0.066 M Na$_2$HPO$_4$ (pH 3.3)) to remove noncovalent protein-protein interactions between TMPs and the surface of DCs. The treatment was stopped by adding an excess amount of 5% FBS in PBS. After washing, the cells were stained with TCRβ for flow cytometry. For DC activation, TMP transfer was allowed to proceed for 24 h before harvesting. Alternatively, purified 4-TMPs were added to culture medium containing DCs in the upper compartment and incubated for 2 h to 24 h before harvesting. For the in vivo assay, OTII TCR transgenic mice were administered OVA257-264 or OVA323-339 (100 μg; intraperitoneally (i.p.)) in PBS, respectively. After 48 h, draining lymph nodes were taken and dissociated.

**Electron microscopy**. For SEM, cells were fixed with 2.5% glutaraldehyde solution for 2 h, rinsed with PBS for 5 min, and fixed in osmium tetroxide for 2 h. Samples were then dehydrated in a graded ethanol series over 30 min and dried in a critical point dryer. Samples were prepared by sputter coating with 1 nm to 2 nm gold-palladium and analysed using field-emission SEM (Hitachi, Tokyo, Japan). For TEM, cells were fixed in suspension at 37 °C in a 5% CO$_2$ incubator. Cells (1 mL) were added to 9 mL of fixative (2.2% glutaraldehyde in 100 mM NaPO$_4$ (pH 7.4)) and fixed for 2 h. Cells were post-fixed in 1% osmium tetroxide, stained en-bloc with 0.5% uranyl acetate in water, dehydrated in a graded ethanol series, embedded, and thin-sectioned. Sections were stained with 2% uranyl acetate in methanol for 20 min, followed by lead citrate for 5 min, and then observed using a Tecnai 12 electron microscope (FEI, Hillsboro, OR, USA) at 120 kV under low-dose conditions.

**Statistics**. Student's $t$ test and one-way analysis of variance (corrected for all pairwise comparisons) were performed using Prism software. A $P < 0.05$ was considered statistically significant.

## Data availability

The data that support the findings of this study are available from the corresponding author upon request. Mass spectra are available under MassIVE MSV000082822 and ProteomeXchange PXD010789.

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

## Acknowledgements

This work was supported by the Creative Research Initiative Program (2015R1A3A2066253) and the Basic Science Program (2015R1A2A1A15052658) through National Research Foundation (NRF) grants funded by the Ministry of Science, ICT & Future Planning (MSIP), and Basic Science Program (2013R1A6A3A04064259) through National Research Foundation (NRF) grants funded by the Ministry of Education (MOE), Korea, and the GIST Research Institute (GRI) in 2017. We would like to thank Dr. Jin Young Kim and Dr. Ju Yeon Lee (Korea Basic Science Institute, Ochang Headquarter, Division of Bioconvergence Analysis) for the nanoLC-LTQ-Orbitrap analysis.

## Author contributions

H-R.K. conceived the study; H-R.K., Y.M. and K-S.L. designed and performed the experiments; Y-J.P., J-S.P., J-H.P., and B-N.J. performed the experiments and analysed data; C-H.K. and Y-M.H. prepared the essential reagents; M.K., Y.J., S-M.L., C-S.P. and S-H.I. assisted in the experiments and wrote the manuscript; H-R.K. and C-D.J. wrote the manuscript. All of the authors edited the manuscript.

## Additional information

**Competing interests:** The authors declare no competing interests.

