## [Peer Review File · Nature Communications]

Reviewers' comments:

Reviewer #1 (Immunological synapse, TCR signalling)(Remarks to the Author):

In the present work, H.R Kim et al. used a combination of techniques (including: electron microscopy, confocal live cell imaging, total internal reflection microscopy, isolation and purification of T cell microvilli particles, liquid chromatography-mass spectrometry) to investigate the role of T cell microvilli in conveying activation signals to antigen-presenting cells (APC). They report that following TCR-mediated triggering T cells release microvilli-derived membrane particles (T-cell microvilli particles, TMPs). They also report that TMPs from CD4+ T cells contain several molecules, including TCR/CD3, LFA-1 and CD2 and can induce dendritic cell activation regardless of TCR engagement.

The present study is interesting and is based on an impressive amount of results. Images are gorgeous.

However, some problems diminish my enthusiasm for the present manuscript. I am also concerned about the scope of the study and the relevance of the reported findings for T cell/APC biology.

Specific points

1) The manuscript is difficult to read and contains several typos that make it difficult to fully appreciate the main message of the study.

The entire text should be amended. It would be important to clarify the biological meaning of the reported findings.

2) Results presented in figure Fig. 8 are not convincing. The Figure shows that TMPs released by OT-II CD4+ T cells on lipid bilayer activate DC even if they are not loaded with specific antigenic peptide. However, the TMPs were generated on lipid bilayer that contained specific pMHC. Those pMHC might be present in the TMPs preparation and might bind APC. It also is important to exclude the possibility that a few T cells might be present in the TMPs preparation and therefore be responsible of the observed DC responses. Intriguingly, Fig 8A depicts a DC undergoing a significant $[Ca^{2+}]_i$ increase that appears to be superimposed to a small round cell that resembles a T cell.

3) Data in Figure 3 C and D are not fully convincing. It is not clear why TCR+ vesicles released during T cell locomotion (red dots) loose V5G staining. Could the detected vesicle be (at least in part) released via mechanisms independent from microvilli formation?

4) The biological relevance of the reported results in a physiological context is not clear. Are MTVs important to selectively activate cognate APCs or are they involved in bystander APC activation? The activation of cognate APC occurs within stable IS via a variety of cell-cell signals (CD40/CD40L; cytokines, adhesion molecules etc). It is not clear what would be the advantage of MTVs release over the formation of a stable IS in cognate APC activation. Alternatively, it is not clear how, MTVs released by a given T cell upon the encounter with a cognate APC might move to bystander APCs and trigger those cells.

Reviewer #2 (TCR signaling, phosphoproteomic)(Remarks to the Author):

This manuscript by Chang-Duk Jun and co-workers uses electron scanning microscopy (ESM) and fluorescence imaging to describe morphological features displayed by normal mouse T cells when stimulated with TCR ligands. They focus on microvilli morphology and dynamics before and after T

cells engagement with either antigen-pulsed APCs or plastic-coated activating Abs (anti-CD3 + anti-CD28) or artificial lipid bilayers presenting pMHC and integrin ligand.

Recent data suggest that the tip of the microvilli may contain segregated TCR (or TCR clusters) ready to engage with ligands on APCs. T cell microvilli are rapidly becoming an attractive subject of investigation for the potential to explain some biological features of T cells. However, it is still unclear if such structures are really present in vivo on resting T cells scanning APCs, if they are also stimulated by experimental manipulations (e.g., PLL immobilization, known to induce signalling perturbation of resting T cells) and/or are the consequence of TCR recognition and initial cell activation rather than being constitutively present on T cells.

An important technical advance of the present work is the expression in T cells of GFP-VstM5f, a protein that localizes at and close to the tip of microvilli observed by ESM. This approach renders possible to follow in time and space how the microvilli extend and change in their morphology when a T cell is fully activated. The work describes that the short microvilli covering the T cell surface are seen as extended long and thin digitations onto the APC (or solid material or bilayer) surface when cognate antigen is offered. They also describe that these long membrane extensions leave behind membrane material. This phenomenon is suggested to be originated by a membrane-modification process called "troglodytosis". Using a number of rather convincing evidence, the authors conclude that the material left behind by activating T cells is derived from microvilli and contains proteins involved in membrane budding. That in vitro stimulated T cells leave membrane material on APCs has been reported already more than twenty years ago. However, it is not yet clear at present if and to what extent such a morphological modification is also happening in vivo where cognate pMHC is likely to be very scarce. Also, there is no evidence as yet as to whether this phenomenon has biological relevance. This idea is resumed by the authors as "immunological synaptosomes that carry T cell messages to APCs" However, what is the message delivered to the APC is yet unknown.

No doubt that the authors have generated a potent experimental tool to study microvilli dynamics in T cells and made a very accurate and detailed description of these phenomena. However, the main message, if there is really a precise one, is lost in a large number of experiments ranging from trying to prove the use of a valid microvilli marker; the presence of TCR-zeta at the tip of the microvilli; define "phases " of microvilli genesis and termination; the onset of left-over budding vesicles and how they seem to form; their dependence on LFA1; their protein content using MS and a supposed activation of CDs without TCR engagement, an apparent odd conclusion if one consider that the process is initiated by TCR engagement on the APC. As presently conceived and structured in writing, this work does not convey any solid comprehensive knowledge on microvilli biology in T cells nor clarify a possible biological role of TMPs

In conclusion, my main concern is that as the work is purely phenomenological and provides only limited advancement as to what is the real functional meaning of the microvilli on T cells. We are left with guesses and no firm conclusion on almost every question addressed by this work. A strong suggestion is to focus on one or two aspects and provide more firm conclusions so to benefit the scientific community with additional solid knowledge to guide towards a better understanding the role of microvilli in T cells

Reviewer #3 (TCR signaling, costimulation)(Remarks to the Author):

The manuscript by Kim and co-authors analyzed the function of T cell microvilli on antigen-presenting cells (APC). Since there exists little information about the microvilli function and its effects on APCs,

this manuscript aims to characterize the presence of T cell proteins effecting APC activity in microvilli and their release in order to modulate APC function. They demonstrate that these microvilli carry T cell receptors (TCR), which are released and contain particles including proteins involved in the activation of dendritic cells independent of the TCR engagement suggesting that these T cell microvilli are immunological synaptosomes, which give their messages to APCs. This is a hot and timely topic in particular concerning the role of trogocytosis in immune response, which could be bidirectional demonstrating that APC could acquire the T cell receptor, while T cells could acquire also peptide bound MHC molecules.

The experiments were carefully performed using state of the art technologies. There are only some minor issues, which have to be address by the authors:

- The release of composition of T cell microvilli particles (TMP) and their involvement in APC activity and T cell interaction was investigated.
- In Figure 6 the statistical analysis is missing. How often were the experiments performed?
- Figure 6C a marker such as β -actin is missing to show similar loading. Similar holds for Figure 7D.
- Analysis of Vps4 is missing.

Response to Reviewer #1:

Dear Reviewer,

We appreciate your insightful and critical comments regarding our manuscript entitled, “**T-cell microvilli constitute immunological synaptosomes that carry messages to antigen-presenting cells (NCOMMS-18-07992)**”. In response to your comments, my colleagues and I have made significant changes to the manuscript, and we have addressed your comments, point by point. Our responses and accompanying revisions are as follows:

Introductory and general comments from Reviewer #1.

In the present work, H.R Kim et al. used a combination of techniques (including: electron microscopy, confocal live cell imaging, total internal reflection microscopy, isolation and purification of T cell microvilli particles, liquid chromatography-mass spectrometry) to investigate the role of T cell microvilli in conveying activation signals to antigen-presenting cells (APC). They report that following TCR-mediated triggering T cells release microvilli-derived membrane particles (T-cell microvilli particles, TMPs). They also report that TMPs from CD4+ T cells contain several molecules, including TCR/CD3, LFA-1 and CD2 and can induce dendritic cell activation regardless of TCR engagement. The present study is interesting and is based on an impressive amount of results. Images are gorgeous. However, some problems diminish my enthusiasm for the present manuscript. I am also concerned about the scope of the study and the relevance of the reported findings for T cell/APC biology.

< Specific points >

Comment 1) The manuscript is difficult to read and contains several typos that make it difficult to fully appreciate the main message of the study. The entire text should be amended. It would be important to clarify the biological meaning of the reported findings.

Answer → Thank you for your constructive and helpful comments. English is a second language for me and it is not always easy to convey clearly my original intent. In this revised manuscript, we have tried to clarify the reasons behind our study and the main physiological meaning of our findings. In addition, we performed new experiments and have included them in the revised manuscript (Text and Figures). The revised sections are highlighted with yellow color in the text.

Moreover, although the first edition of our manuscript was edited by a professional editing service (Editage company, Cactus Communications, Inc.), we asked them to re-edit our manuscript for further improvement. Please find attached the certificate letter. Editage is a global scientific communication company offering editing, translation, medical writing, and other services to researchers.

Comment 2) Results presented in Fig. 8 are not convincing. The Figure shows that TMPs released by OT-II CD4+ T cells on lipid bilayer activate DC even if they are not loaded with specific antigenic peptide. However, the TMPs were generated on lipid bilayer that contained specific pMHC. 1) Those pMHC might be present in the TMPs preparation and might bind APC. 2) It also is important to exclude the possibility that a few T cells might be present in the TMPs preparation and therefore be responsible of the observed DC responses

Answer → Thank you for your insightful comments.

Answer for question 1) As you mentioned, because 4-TMPs in Figure 8A were from OT-II CD4⁺ T cells on lipid bilayer presenting peptide-MHC and ICAM-1, we cannot rule out the contamination of pMHC in the TMPs preparation. However, because 4-TMPs purified from CD4⁺ T cells on anti-CD3-coated surface also activated the DCs, we have no doubt that 4-TMPs are able to activate DCs in a cognate antigen-independent manner. As I have mentioned in the manuscript or 'General messages to reviewer #1', there are many reports in the literature showing that MHC class II signaling does not appear to play a role in DC activation, while antigen-specific T-cell interaction may contribute via other mechanisms to regulate DC activation.

Answer for question 2) As you can see in Figure S7A and B, we provided the unmerged, separated images to exclude any suspicions of the T-cell contamination in the TMP preparations. In addition, we added the results of calcium response measured in many cells to increase the credibility of our results (Figure S7C).

Many pieces of evidence suggest that extracellular vesicles can regulate the activity of DCs or interacting cells. Although TMPs are produced by TCR activation by recognizing cognate antigen-MHC, DC can be activated via other mechanisms such as membrane proteins on or cytokines from T cells rather than MHC signals. In fact, it is well known that DC can also be activated by innate immunity signals such as TLR signals.

Comment 3) Data in Figure 3C and D are not fully convincing. It is not clear why TCR⁺ vesicles released during T cell locomotion (red dots) lose V5G staining. Could the detected vesicle be (at least in part) released via mechanisms independent from microvilli formation?

Answer and Revised → V5G is a GFP construct. It can be lost by quenching due to laser exposure, over time, but it does not indicate the disappearance of TMPs. As you can see in the results, most GFP (V5G) signals are always co-localized with red (TCR β) signals during synapse formation (Figure 3A). We only see the gradual disappearance of V5G signals after separation from T cells in kinapses. However, to clearly reveal the presence of V5G signals, we also provided the unmerged, separated images of Figure 3D and Movie S4 in Figure S4B (arrowheads).

A previous report by Choudhuri *et al* (Nature 2014, 507:118) demonstrated that 'TCR-enriched microvesicles' are generated from cSMAC of stable IS. However, if TCRs mostly pre-exist at the microvilli tips, we can imagine that TCR-enriched vesicles are released from the microvilli. Moreover, we presented here that, in contrast to the previous report (J Immunol 173, 4985, 2004), microvilli do not disappear under activated conditions, rather, they may be covered by lamellipodial protrusions that result from active actin polymerization at the dSMAC. As shown in Figure S3B (orthogonal views), V5G signals are always co-localized with the F-actin signals (F-actin spots) as indicated by white arrowheads. This result implies that F-actin bundles, which originally constitute the microvilli, do not disappear during IS formation.

A potential mechanism for how microvilli were likely disappeared from the dSMAC, moved towards the cSMAC with TCR clusters during IS formation and reappeared after synapse breaking is schematically depicted here to help your understanding.

Comment 4) The biological relevance of the reported results in a physiological context is not clear. Are TMPs important to selectively activate cognate APCs or are they involved in bystander APC activation? The activation of cognate APC occurs within stable IS via a variety of cell-cell signals (CD40/CD40L; cytokines, adhesion molecules etc).

It is not clear what would be the advantage of TMPs release over the formation of a stable IS in cognate APC activation. Alternatively, it is not clear how TMPs released by a given T cell upon the encounter with a cognate APC might move to bystander APCs and trigger those cells.

Answer and Revised → Thank you for your very important comments. According to your comments, we performed the experiments as depicted in Figure 9A and B. In Figure 9A, TCR β^+ and V5G $^+$ signals were only observed on the surface of DCs incubated with OTII CD4 $^+$ T cells in the presence of OVA peptide (d = 19.4 and 18.6% of TCR β^+ and V5G $^+$ DCs, respectively) but not on DCs with other conditions, suggesting that TMPs are only released under the condition that T cells interact with cognate antigen-bearing DCs. Moreover, in instances where T did not physically interact, DCs were not activated in terms of surface expression of MHC class II, CD80, CD86, and CD40 (Figure 9A). These results strongly suggest that TMPs are important for selectively activating cognate DCs. Similar results were obtained with the purified 4-TMPs as described in Figure 9B.

However, as we mentioned earlier, the nature of TMPs to activate DCs was not dependent on the cognate antigen and TCR, suggesting that TMPs activate DCs via other mechanisms such as CD2 and LFA-1. In this revised version, we further observed that 4-TMPs are enriched with cytokines or secreting proteins that are already known to regulate DC activations (See Figure 10C). These results demonstrate that not a single but several factors in the TMPs could be involved in the activation of DCs

Interestingly, a recent report demonstrated that all major T cell subsets spend more time in the kinapse mode and that durable interactions for priming of T cells do not require stable synapses (Cell Reports 22, 340, 2018). This report suggests that DC activation also does not require stable synapses, but instead, it may require TMPs as they contain a large number of messages for DC activation. Consistent with our hypothesis, we found that T cells produce more TMPs in the kinapse mode than in the synapse mode (Figure 3C).

<General Messages to Reviewer #1>

Although T cells are known to release slightly different types of extracellular vesicles (EVs), these EVs can be classified as exosomes and ectosomes according to their subcellular origin. Exosomes are known to be formed by inward budding of the limiting membrane of the late endosome, whereas ectosomes bud directly from the plasma membrane. However, many previous reports regarding exosomes in T cells may have been biased in collecting exosome particles, as researchers usually used an anti-CD3 antibody-coated surface to activate T cells. For example, Blanchard *et al.* (J Immunol 2002, 168:3235) demonstrated that TCR activation of human T cells induces the production of exosomes bearing the TCR/CD3/ ζ complex. According to our current results, Rab11⁺ intracellular vesicles are free of TCR/CD3/ ζ complexes. In contrast, particles derived from microvilli were highly enriched with TCR complex as well as other TCR-related proteins. In fact, through the current investigation, we now understand that T cells release significant amounts of TCR-enriched, T-cell microvilli particles (TMPs) on the surface of anti-CD3-coated coverglasses or lipid bilayers presenting peptide-MHC and ICAM-1. However, treatment of T cells with anti-CD3 or PMA/ionophore does not produce any particles from T cells, suggesting that adhesion to the surface, in addition to the TCR activation, is a critical mechanism to produce TMPs (or microvesicles). Interestingly, Blanchard *et al.* also demonstrated that PMA/ionophores do not induce exosome release. Furthermore, the proteins they discovered in the exosomes were very similar to those discovered in the TMPs (see Fig. 6C). These results suggest that the EVs purified by Blanchard *et al.* are not exosomes but rather are presumably derived from TCR-enriched, T-cell microvillus particles, i.e., TMPs.

'TCR-enriched microvesicles' are another example of the different types of EVs that were reported by Choudhuri *et al.* (Nature 2014, 507:118), who discovered that TCR-enriched microvesicles are released from the central supramolecular activation clusters (cSMAC) of immunological synapses (IS) by membrane budding processes. At that time, however, they had no information about the 'secretory domain' in which the TCR-enriched microvesicles are released. However, we now understand that microvilli serve as the structural scaffold for clustering of TCRs and for the recruitment of microclusters upon TCR engagement (PNAS 2016, 113:E5916). Using GFP-tagged probes, specific for microvilli, we also noticed that TCRs are obviously clustered at the microvillus tips. Our present results, in addition to the previous report by Jung *et al.* (PNAS 2016, 113:E5916), naturally raises the question of whether microvilli can bring TCRs to the cSMAC through centripetal actin flow. We unambiguously saw that microvilli signals moved together with TCR clusters in ISs as well as T-cell kinapses. Further, we found that TCR clusters were separated from the T-cell body when microvilli were detached by the process of trogocytosis. Taken together, all the currently available data strongly demonstrate that both TCR-enriched microvesicles (Nature 2014, 507:118) and exosomes (J Immunol 2002, 168:3235) are of the same origin: microvilli.

In addition to identifying the cellular origin of microparticles, we discovered here that microvilli have more diverse biological functions rather than merely acting as sensory or adhesion organs. Microvillus tips are concentrated with direct membrane budding complexes that induce further fragmentation of large microvillus particles to the small, exosome-sized TMPs (S-TMPs). We found that small 4-TMPs (derived from CD4⁺ T cells) are only transferred to the cognate antigen-bearing DCs during physical interaction but to the bystander DCs. However, contrary to the point raised by you and reviewer #2, DC activation was independent of cognate antigen and TCR presented on TMPs. Indeed, there are many reports in the literature showing that MHC class II signaling does not appear to play a role in DC activation, while antigen-specific T-cell interaction may contribute via other mechanisms to regulate DC activation (so far, no single mechanism has been identified). Thus, we believe that not a single factor, but many factors in TMPs (or T cells) can contribute to the activation of DCs. For example, in addition to the CD2 and LFA-1 presented in the first submission, we newly found that

cytokines or secreting proteins which are known to control the activation or maturation of DCs are enriched in the TMPs.

During our current work, we also recognized that we need to change the current paradigm that we have understood from the studies of IS using artificial lipid bilayer systems. In many previous studies, researchers observed that TCR nanoclusters or microclusters are newly formed on the flat 2-dimensional membranes and thought that they are accumulated into the cSMAC via actin flow. However, the results of this study suggest that TCRs are in microvilli tips and their movements are associated with the movement of microvilli on the surface of T cells. In addition, we must reconsider the reason why TCRs almost disappear from the cell membrane after T cell activation. This phenomenon may be caused by the fact that microvilli are dropped rather than internalized in T cells. Microvilli particles separated from the T cell body can attach to the surface of DCs in the form of vesicles, which not only may prolong the activity of the DCs but also may result in the partial introduction of the T cell trait into the DCs.

We believe that our results clearly define the role of T-cell microvilli and microvilli particles, i.e., TMPs, which had previously been ambiguous in terms of their cellular source. Thus, the present results are novel and make significant conceptual advances in the field of T cell immunology, especially the communications among immune cells.

Response to Reviewer #2:

Dear Reviewer,

We appreciate your insightful and critical comments regarding our manuscript entitled, “**T-cell microvilli constitute immunological synaptosomes that carry messages to antigen-presenting cells (NCOMMS-18-07992)**”. In response to your comments, my colleagues and I have made significant changes to the manuscript, and we have addressed your comments, point by point. Our responses and accompanying revisions are as follows:

Introductory and general comments from Reviewer #2.

This manuscript by Chang-Duk Jun and co-workers uses electron scanning microscopy (ESM) and fluorescence imaging to describe morphological features displayed by normal mouse T cells when stimulated with TCR ligands. They focus on microvilli morphology and dynamics before and after T cells engagement with either antigen-pulsed APCs or plastic-coated activating Abs (anti-CD3 + anti-CD28) or artificial lipid bilayers presenting pMHC and integrin ligand. Recent data suggest that the tip of the microvilli may contain segregated TCR (or TCR clusters) ready to engage with ligands on APCs. T cell microvilli are rapidly becoming an attractive subject of investigation for the potential to explain some biological features of T cells. **However, it is still unclear if such structures are really present in vivo on resting T cells scanning APCs, if they are also stimulated by experimental manipulations (e.g., PLL immobilization, known to induce signaling perturbation of resting T cells) and/or are the consequence of TCR recognition and initial cell activation rather than being constitutively present on T cells.** An important technical advance of the present work is the expression in T cells of GFP-Vstm5, a protein that localizes at and close to the tip of microvilli observed by ESM. This approach renders possible to follow in time and space how the microvilli extend and change in their morphology when a T cells is fully activated. The work describes that the short microvilli covering the T cell surface are seen as extended long and thin digitations onto the APC (or solid material or bilayer) surface when cognate antigen is offered. They also describe that these long membrane extensions leave behind membrane material. This phenomenon is suggested to be originated by a membrane-modification process called “trogocytosis”. Using a number of rather convincing evidence, the authors conclude that the material left behind by activating T cells is derived from microvilli and contains proteins involved in membrane budding. That in vitro stimulated T cells leave membrane material on APCs has been reported already more than twenty years ago. **However, it is not yet clear at present if and to what extent such a morphological modification is also happening in vivo where cognate pMHC is likely to be very scarce. Also, there is no evidence as yet as to whether this phenomenon has biological relevance. This idea is resumed by the authors as “immunological synaptosomes that carry T cell messages to APCs” However, what is the message delivered to the APC is yet unknown.**

Answer and Revised → Thank you for your very critical comments. In fact, the questions raised are really difficult to be answered because, to our knowledge, there is no report defining the surface structures of T cells during dynamic interaction with APCs *in vivo*. However, the Vstm5 protein, which we discovered in this report, is very specific for microvilli, and hence we are establishing a T-cell specific Vstm5_GFP (V5G) transgenic mice. Using two-photon intravital microscopy, we are going to unveil the dynamic behaviors of microvilli *in vivo*. We hope we can measure how T cells appear *in vivo* before and after activation with cognate DCs. Unfortunately, however, the project is still ongoing and therefore we are very sorry we cannot provide the data in this revised version.

However, as you can see in the figure below, scanning of the surface of the lipid bilayer presenting peptide-MHC and ICAM-1 showed that the T cells in kinapses have multiple microvilli-like membrane protrusions (arrowheads). This situation may be very similar to the condition that T cells interact with APCs *in vivo*. Just as we believe the centripetal accumulation of TCR clusters on lipid bilayer, we also believe the formation of microvilli tails and microvilli-originated particles when T cells are in kinapses.

T cells on lipid bilayer presenting pMHC and ICAM-1 (**kinapses**)

T cells on lipid bilayer presenting pMHC and ICAM-1 (**synapses**)

In addition, in this revised manuscript, we provide the *in vivo* evidence of TMP transfer to the physically interacting DCs (See Figure 9C). Our results suggested that TMPs (or trogocytosis and vesiculation as reported in this paper) are also transferred *in vivo*.

According to the comments from reviewer #1, we also performed additional new experiments as shown in Figure 9 and 10. In Figure 9A, TCR β^+ and V5G $^+$ signals were only observed on the surface of DCs incubated with OTII CD4 $^+$ T cells in the presence of OVA peptide (d = 19.4 and 18.6% of TCR β^+ and V5G $^+$ DCs, respectively) but not on DCs with other conditions, suggesting that TMPs are only released under the condition that T cells interact with cognate antigen-bearing DCs. Moreover, instances where T cells did not physically interact, DCs were not activated in terms of surface expression of MHC class II, CD80, CD86, and CD40 (Figure 9A). These results strongly suggest that TMPs are important to selectively activate cognate DCs. Similar results were obtained with the purified 4-TMPs as described in Figure 9B.

However, as we mentioned earlier, the nature of TMPs to activate DCs was not dependent on the cognate antigen and TCR, suggesting that TMPs activate DCs via other mechanisms such as CD2 and LFA-1. In this revised version, we further observed that 4-TMPs are enriched with the cytokines or secreting proteins that are already known to regulate DC activations (See Figure 10C). These results demonstrate that not a single but several factors in the TMPs could be involved in the activation of DCs

Interestingly, a recent report demonstrated that all major T cell subsets spend more time in the kinapse mode, and durable interactions for priming of T cells do not require stable synapses (Cell Reports 22, 340, 2018). This report suggests that DC activation also does not require stable synapses, but instead, it may require TMPs as they contain a large number of messages for DC activation. In agreement with our hypothesis, we found that T cells produce more TMPs in the kinapses mode than in the synapses

mode (Figure 3C).

Comment 1) No doubt that the authors have generated a potent experimental tool to study microvilli dynamics in T cells and made a very accurate and detailed description of these phenomena. However, the main message, if there is really a precise one, is lost in a large number of experiments ranging from trying to prove the use of a valid microvilli marker; the presence of TCR-zeta at the tip of the genesis and termination; the onset of left-over budding vesicles and how they seem to form; their dependence on LFA1; their protein content using MS and a supposed activation of DCs without TCR engagement, **an apparent odd conclusion if one consider that the process is initiated by TCR engagement on the APC**. As presently conceived and structured in writing, this work does not convey any solid comprehensive knowledge on microvilli biology in T cells nor clarify a possible biological role of TMPs

Answer and Revised → Thank you for your constructive and helpful comments. English is a second language for me and it is not always easy to convey clearly my original intent. In this revised manuscript, we tried we have tried to clarify the reasons behind our study and the main physiological meaning of our findings. In addition, we performed new experiments and have included them in the revised manuscript to better explain the function of TMPs in the context of immunological responses (Text and Figures).

Moreover, although the first edition of our manuscript was edited by a professional editing service (Editage company, Cactus Communications, Inc.), we asked them to re-edit our manuscript for further improvement. Please find attached the certificate letter. Editage is a global scientific communication company offering editing, translation, medical writing, and other services to researchers.

On the other hand, I suppose our messages are not unclear in this paper. From our current results, we are convinced that T-cell activation and the resulting TMP generation strictly require cognate antigen MHC–TCR interaction. However, DC activation by activated T cell is independent of TCR engagement. Indeed, there are many reports in the literature showing that MHC class II signaling does not appear to play a role in DC activation, while antigen-specific T-cell interaction may contribute via other mechanisms to regulate DC activation. In contrast to the DC activation, there are several reports that B cell activation requires contact-mediated signals from T cell and the first of such signals to be identified is MHC class II, demonstrating that DCs and B cells are activated in different ways. We discussed this part in the Discussion section (highlighted in yellow color).

Comment 2) In conclusion, my main concern is that as the work is purely phenomenological and provides only limited advancement as to what is the real functional meaning of the microvilli on T cells. We are left with guesses and no firm conclusion on almost every question addressed by this work. A strong suggestion is to focus on one or two aspects and provide more firm conclusions so to benefit the scientific community with additional solid knowledge to guide towards a better understanding the role of microvilli in T cells.

General messages to the reviewer #2 → Although T cells are known to release slightly different types of extracellular vesicles (EVs), these EVs can be classified as exosomes and ectosomes according to their subcellular origin. Exosomes are known to be formed by inward budding of the limiting membrane of the late endosome, whereas ectosomes bud directly from the plasma membrane. However, many previous reports regarding exosomes in T cells may have been biased in collecting exosome particles, as researchers usually used an anti-CD3 antibody-coated surface to activate T cells. For example, Blanchard *et al.* (J Immunol 2002, 168:3235) demonstrated that TCR activation of human T cells induces the production of exosomes bearing the TCR/CD3/ζ complex. According to our

current results, Rab11⁺ intracellular vesicles are free of TCR/CD3/ ζ complexes. In contrast, particles derived from microvilli were highly enriched with TCR complex as well as other TCR-related proteins. In fact, through the current investigation, we now understand that T cells release significant amounts of TCR-enriched, T-cell microvilli particles (TMPs) on the surface of anti-CD3-coated coverglasses or lipid bilayers presenting peptide-MHC and ICAM-1. However, treatment of T cells with anti-CD3 or PMA/ionophore does not produce any particles from T cells, suggesting that adhesion to the surface, in addition to the TCR activation, is a critical mechanism to produce TMPs (or microvesicles). Interestingly, Blanchard *et al.* also demonstrated that PMA/ionophores do not induce exosome release. Furthermore, the proteins they discovered in the exosomes were very similar to those discovered in the TMPs (see Fig. 6C). These results suggest that the EVs purified by Blanchard *et al.* are not exosomes but rather are presumably derived from TCR-enriched, T-cell microvillus particles, i.e., TMPs.

'TCR-enriched microvesicles' are another example of the different types of EVs that were reported by Choudhuri *et al.* (Nature 2014, 507:118), who discovered that TCR-enriched microvesicles are released from the central supramolecular activation clusters (cSMAC) of immunological synapses (IS) by membrane budding processes. At that time, however, they had no information about the 'secretory domain' in which the TCR-enriched microvesicles are released. However, we now understand that microvilli serve as the structural scaffold for clustering of TCRs and for the recruitment of microclusters upon TCR engagement (PNAS 2016, 113:E5916). Using GFP-tagged probes, specific for microvilli, we also noticed that TCRs are obviously clustered at the microvillus tips. Our present results, in addition to the previous report by Jung *et al.* (PNAS 2016, 113:E5916), naturally raises the question of whether microvilli can bring TCRs to the cSMAC through centripetal actin flow. We unambiguously saw that microvilli signals moved together with TCR clusters in immunological synapses as well as T-cell kinapses. Further, we found that TCR clusters were separated from the T-cell body when microvilli were detached by the process of trogocytosis. Taken together, all the currently available data strongly demonstrate that both TCR-enriched microvesicles (Nature 2014, 507:118) and exosomes (J Immunol 2002, 168:3235) are of the same origin: microvilli.

In addition to identifying the cellular origin of microparticles, we discovered here that microvilli have more diverse biological functions rather than merely acting as sensory or adhesion organs. Microvillus tips are concentrated with direct membrane budding complexes that induce further fragmentation of large microvillus particles to the small, exosome-sized TMPs (S-TMPs). We found that small 4-TMPs (derived from CD4⁺ T cells) are only transferred to the cognate antigen-bearing DCs during physical interaction but to the bystander DCs. However, contrary to the point raised by you and reviewer #2, DC activation was independent of cognate antigen and TCR presented on TMPs. Indeed, there are many reports in the literature showing that MHC class II signaling does not appear to play a role in DC activation, while antigen-specific T-cell interaction may contribute via other mechanisms to regulate DC activation (so far, no single mechanism has been identified). Thus, we believe that not a single factor, but many factors in TMPs (or T cells) can contribute to the activation of DCs. For example, in addition to the CD2 and LFA-1 presented in the first submission, we newly found that cytokines or secreting proteins which are known to control the activation or maturation of DCs are enriched in the TMPs.

During our current work, we also recognized that we need to change the current paradigm that we have understood from the studies of IS using artificial lipid bilayer systems. In many previous studies, researchers observed that TCR nanoclusters or microclusters are newly formed on the flat 2-dimensional membranes and thought that they are accumulated into the cSMAC via actin flow. However, the results of this study suggest that TCRs are in microvilli tips and their movements are associated with the movement of microvilli on the surface of T cells. In addition, we must reconsider the reason why TCRs almost disappear from the cell membrane after T cell activation. This

phenomenon may be caused by the fact that microvilli are dropped rather than internalized in T cells. Microvilli particles separated from the T cell body can attach to the surface of DCs in the form of vesicles, which not only may prolong the activity of the DCs but also may result in the partial introduction of the T cell trait into the DCs.

We believe that our results clearly define the role of T-cell microvilli and microvilli particles, i.e., TMPs, which had previously been ambiguous in terms of their cellular source. Thus, the present results are novel and make significant conceptual advances in the field of T cell immunology, especially the communications among immune cells. Thank you.

Response to Reviewer #3:

Dear Reviewer,

We appreciate your insightful and critical comments regarding our manuscript entitled, “**T-cell microvilli constitute immunological synaptosomes that carry messages to antigen-presenting cells (NCOMMS-18-07992)**”. In response to your comments, my colleagues and I have made significant changes to the manuscript, and we have addressed your comments, point by point. Our responses and accompanying revisions are as follows:

Introductory and general comments from Reviewer #3.

The manuscript by Kim and co-authors analyzed the function of T cell microvilli on antigen-presenting cells (APC). Since there exists little information about the microvilli function and its effects on APCs, this manuscript aims to characterize the presence of T cell proteins effecting APC activity in microvilli and their release in order to modulate APC function. They demonstrate that these microvilli carry T cell receptors (TCR), which are released and contain particles including proteins involved in the activation of dendritic cells independent of the TCR engagement suggesting that these T cell microvilli are immunological synaptosomes, which give their messages to APCs. This is a hot and timely topic in particular concerning the role of trogocytosis in immune response, which could be bidirectional demonstrating that APC could acquire the T cell receptor, while T cells could acquire also peptide bound MHC molecules.

The experiments were carefully performed using state of the art technologies. There are only some minor issues, which have to be address by the authors:

Comment 1) In Figure 6 the statistical analysis is missing. How often were the experiments performed?

Answer → We appreciate your insightful comment. As we briefly stated in the Results section, three biological (or experimental) replicates of total, membrane, vesicle, and TMP were prepared and analyzed by LC-MS/MS. After that, the proteins abundantly identified in all three LC-MS/MS analyses were subjected to further data analysis. To show differences in T-cell protein abundance between TMP and vesicle samples, the number of peptide spectral matches (PSMs) representing relative abundance of the identified proteins were compared (Figure 6B). Statistical analysis of LC-MS/MS results was not performed here because relatively wide variations in the number of PSM were observed among the three biological (or experimental) replicates.

Comment 2) Figure 6C a marker such as β -actin is missing to show similar loading. Similar holds for Figure 7D.

Answer and Revised → In Figure 6C, all the western blot results were from same preparations from total, membrane, TMP, and vesicles. The β -actin is shown in the last figure at the right end. We added the β -actin for Figure 7D. Thank you.

Comment 3) Analysis of Vps4 is missing.

Answer and Revised → Analysis of Vps4 (Figure 7F) was shown in Figure 7G in the first submission. See Figure 7G.

REVIEWERS' COMMENTS:

Reviewer #1 (Remarks to the Author):

The authors adequately addressed the point I raised. The revised manuscript is substantially improved. This is an interesting and well-performed study.

Reviewer #2 (Remarks to the Author):

I have carefully reviewed the revised version of this manuscript. However, I have found no substantial improvement that addresses the critics raised in my previous report.

The same limitations previously pointed out subsist. Thus, while the work contains several potentially interesting observations, they are scattered with not great connection with each other. Above all, none of them is really and exhaustively demonstrated. Many questions have been addressed though almost none has been translated to a solidly substantiate new notion.

In the interest of making this extensive, technologically-rich, multi-faceted and expensive investigation a more substantial and effective contribution to the understanding of microvilli function in T cell biology, I strongly suggest to the authors to consider to split it and write an initial manuscript that paves the way to a more substantial work with perhaps in vivo evidence for TMPs using a tg mouse model already in the making.

In a first publication, they could define for instance more substantially the new notion that vesicles produced by T cells are derived from budding structures, connect this better experimentally to microvilli and illustrate more extensively the vesicles' content, notions that are really appealing for T cell biologists. It would be for instance exciting to ask if activated cytotoxic effector T cells use kinapses to deposit on target cells vesicles containing lytic granules. Or effector CD4 T cells use a similar mechanism to leave activating cytokines-containing vesicles on DCs and macrophages.

Reviewer #3 (Remarks to the Author):

The article by Kim and co-authors is a revised version. The authors did a number of important changes throughout the manuscript, which improved it a lot. This includes information concerning trogocytosis and microvilli in particular also the endosomal sorting complex required for transport. They included novel results, changed also the figures and added statistics. This includes in particular regarding the transfer of membrane T cell proteins to the cell surface of dendritic cells, the physical contact of TMP and DC for transfer of TMPs into DC in vivo. Furthermore, using blocking antibodies the authors demonstrated that not only co-stimulatory molecules but also other factors, which still have to be identified, are involved in activation of APCs. This is critically and in depth discussed by the authors. Furthermore, the authors suggest that TMPs present another class of extracellular vesicles. Based on the integration of the new data, which were presented in novel figures and their critical discussion in the view of known literature improved the paper a lot. However, as stated by the authors, there is still urgent need to understand the mechanisms of TMP internalization leading to DC activation and its potential use in therapeutics.

Reviewer #1 (Remarks to the Author):

The authors adequately addressed the point I raised. The revised manuscript is substantially improved. This is an interesting and well-performed study.

Answer → Thank you again for your kind and very constructive review.

Reviewer #2 (Remarks to the Author):

I have carefully reviewed the revised version of this manuscript.

However, I have found no substantial improvement that addresses the critics raised in my previous report.

The same limitations previously pointed out subsist. Thus, while the work contains several potentially interesting observations, they are scattered with not great connection with each other. Above all, none of them is really and exhaustively demonstrated. Many questions have been addressed though almost none has been translated to a solidly substantiate new notion.

In the interest of making this extensive, technologically-rich, multi-faceted and expensive investigation a more substantial and effective contribution to the understanding of microvilli function in T cell biology, I strongly suggest to the authors to consider to split it and write an initial manuscript that paves the way to a more substantial work with perhaps in vivo evidence for TMPs using a tg mouse model already in the making.

In a first publication, they could define for instance more substantially the new notion that vesicles produced by T cells are derived from budding structures, connect this better experimentally to microvilli and illustrate more extensively the vesicles' content, notions that are really appealing for T cell biologists. It would be for instance exciting to ask if activated cytotoxic effector T cells use kinapses to deposit on target cells vesicles containing lytic granules. Or effector CD4 T cells use a similar mechanism to leave activating cytokines-containing vesicles on DCs and macrophages.

Answer → Thank you again for your constructive and sharp points. It may be helpful if I explain some of the history of this paper. Approximately 4 years ago (in 2014), our group first discovered that TCRs are clustered on microvilli tips, as evidenced by both confocal and TIRF microscopy. Unfortunately, however, another group published very similar results before us in PNAS (Jung, Y. et al, 2016). We were quite frustrated because, at that time, we had also written two manuscripts based on these results. Consequently, we then decided to merge these two stories into one paper, which is the present manuscript. Therefore, if we follow your suggestions, it will result in further delays in reporting these results. I hope you can understand our situation. Thank you.

Reviewer #3 (Remarks to the Author):

The article by Kim and co-authors is a revised version. The authors did a number of important changes throughout the manuscript, which improved it a lot. This includes information concerning trogocytosis and microvilli in particular also the endosomal sorting complex required for transport. They included novel results, changed also the figures and added statistics. This includes in particular regarding the transfer of membrane T cell proteins to the cell surface of dendritic cells, the physical contact of TMP and DC for transfer of TMPs into DC in vivo. Furthermore, using blocking antibodies the authors demonstrated that not only co-stimulatory molecules but also other factors, which still have to be identified, are involved in activation of APCs. This is critically and in depth discussed by the authors. Furthermore, the authors suggest that TMPs present another class of extracellular vesicles. Based on the integration of the new data, which were presented in novel figures and their critical discussion in the view of known literature improved the paper a lot. However, as stated by the authors,

there is still urgent need to understand the mechanisms of TMP internalization leading to DC activation and its potential use in therapeutics.

Answer → Thank you again for your kind and very constructive review. Elucidation of the mechanism of TMP transfer onto DCs is our first priority. We are also currently investigating whether TMP can be applicable for anti-cancer therapy. We look forward to getting interesting results soon.